# Sustainable Water Harvesting for Improving Food Security and Livelihoods of Smallholders under Different Climatic Conditions of India

Pankaj Panwar [1], Deepesh Machiwal [2,*] , Vandita Kumari [2], Sanjay Kumar [3], Pradeep Dogra [1], S. Manivannan [4,5], P. R. Bhatnagar [6], J. M. S. Tomar [7], Rajesh Kaushal [7], Dinesh Jinger [8] , Pradip Kumar Sarkar [9], L. K. Baishya [10], Ningthoujam Peetambari Devi [11], Vijaysinha Kakade [12], Gaurav Singh [7] , Nongmaithem Raju Singh [11], S. Gojendro Singh [11], Abhishek Patel [13] , P. S. Renjith [13], Sharmistha Pal [1], V. K. Bhatt [1], N. K. Sharma [1], O. P. S. Khola [1], Sheetal K. Radhakrishnan [13], V. Kasthuri Thilagam [14], P. L. Bhutia [10], Kouberi Nath [9], Rekha Das [9] , Dhiman Daschaudhuri [9], Arun Kumar [15], G. S. Panwar [15], S. V. Dwivedi [16] , Sanjeev Kumar [3] and B. K. Singh [15]

1. ICAR-Indian Institute of Soil and Water Conservation, Research Centre, Sector 27-A, Chandigarh 160019, Uttar Pradesh, India
2. Division of Natural Resources, ICAR-Central Arid Zone Research Institute, Jodhpur 342003, Rajasthan, India
3. College of Forestry, Banda University of Agriculture & Technology, Banda 210001, Uttar Pradesh, India
4. ICAR-Indian Institute of Soil and Water Conservation, Research Centre, Fern Hill (POST), Udhagamandalam 643004, Tamil Nadu, India
5. ICAR-Central Coastal Agriculture Research Institute, Ela, Old Goa 403402, Goa, India
6. ICAR-Central Soil Salinity Research Institute, Karnal 132001, Haryana, India
7. ICAR-Indian Institute of Soil and Water Conservation, 218 Kaulagarh Road, Dehradun 248195, Uttarakhand, India
8. ICAR-Indian Institute of Soil and Water Conservation, Research Centre, Vasad 334603, Gujarat, India
9. ICAR-Research Complex for North Eastern Hill Region, Tripura Centre, Lembucherra, West Tripura 799210, Tripura, India
10. ICAR-Research Complex for North Eastern Hill Region, Nagaland Centre, Medziphema 797106, Nagaland, India
11. ICAR-Research Complex for North Eastern Hill Region, Umroi Road, Umiam 793103, Meghalaya, India
12. ICAR-National Institute of Abiotic Stress Management, Baramati 413115, Maharashtra, India
13. Regional Research Station, ICAR-Central Arid Zone Research Institute, Bhuj 370105, Gujarat, India
14. ICAR-Sugarcane Breeding Institute, Coimbatore 641007, Tamil Nadu, India
15. College of Agriculture, Banda University of Agriculture & Technology, Banda 210001, Uttar Pradesh, India
16. College of Horticulture, Banda University of Agriculture & Technology, Banda 210001, Uttar Pradesh, India
* Correspondence: deepesh.machiwal@icar.gov.in or dmachiwal@rediffmail.com

**Abstract:** In India, the per capita availability of water is projected to be 1465 m$^3$ and 1235 m$^3$ by the years 2025 and 2050, respectively, and hence, India would be a water-stressed country as per the United Nations' standard of less than 1700 m$^3$ per capita water availability. India is predominantly an agricultural-dominant country. Rainfed agriculture in the country contributes 40% of food grain production and supports half of the human population and two-thirds of the livestock population. The country has 15 different agro-climatic zones, and each agro-climatic region has its own constraints of water availability and management along with the potential for their optimum utilization. Such situations warrant the formulation of regional-level strategies. Efforts were made to integrate and evaluate the feasibility of water harvesting and its utilization at twelve different sites representing six different agro-climatic conditions spanning pan India. It was found that water harvesting through tanks/ponds is a feasible approach and can increase the crop production as well as diversification. The results reveal that the range of crop diversification index increased from 0.49–0.85 to 0.65–0.98; the crop productivity index increased from 0.28–0.66 to 0.66–0.90; the cultivated land utilization index increased from 0.05–0.69 to 0.34–0.84; and the crop water productivity index increased from 0.20–0.51 to 0.56–0.96, among other production and diversification indices, due to additional water availability through rainwater harvesting intervention. Moreover, the gross return increased from INR 43,768–704,356 to INR 220,840–1,469,108 ha$^{-1}$, representing a 108 to 400% increase in the returns due to the availability of water. The findings of this study suggest that the water harvesting in small

ponds/tanks is economical and feasible, requires less technological intervention, and increases crop diversification in all the studied agro-climatic conditions, and hence, the same needs to be encouraged in the rainfed areas of the country.

**Keywords:** water harvesting; crop diversification index; crop production index; crop land utilization index; crop water productivity index

## 1. Introduction

The livelihood of humans revolves around water, from agricultural production to animal husbandry, horticulture, establishment of factories, tourism, etc. [1]. Water is a vital component that determines the full potential of the agriculture sector of a country [2,3]. In the past, water was thought to be in plenty; however, the current scenario is changing because of climate change and the improper utilization of water resources [4]. The per capita freshwater availability in India averages 2214 $m^3$ $year^{-1}$ [5], which is projected to be 1465 and 1235 $m^3$ by the years 2025 and 2050, respectively [6]. According to the United Nations' standard, countries with an annual per capita availability of less than 1700 $m^3$ are considered as water stressed, and those with less than 1000 $m^3$ are considered as water scarce [7,8]. Thus, India would need 2788 billion cubic meters (bcm) of water annually by 2050 to avoid the water stress condition, and 1650 bcm to avoid the water scarcity situation [9]. It was reported that 67% of the country's net cultivated area falls under rainfed agriculture, where crop cultivation depends solely on monsoons [5,7]. Rainfed agriculture in the country contributes to 40% of food grain production and supports half of the human population and two-thirds of the livestock population [10]. The rainfall occurs during monsoon season, and a significant portion of it is escaped from the area as runoff without any utilization in agriculture. Hence, crop productivity drastically reduces due to the non-availability of water, especially at the critical growth stage of the crop [11]. Moreover, moisture stress further affects the nutrient availability to the crop since nutrient mobility depends on optimum soil moisture [12]. India was divided into 15 different agro-climatic zones by the Planning Commission of Government of India based on physiography and climate [13]. Each agro-climate has its own constraints of water availability and management along with the potential for their optimum utilization [14]. Such situations warrant regional-level technologies and strategies to conserve, harvest, and utilize runoff water.

It is expected that the Indian Himalayan region (Figure 1) would be the worst-affected area due to climate change. Sontakke et al. [15] observed a declining rainfall trend for the western Indian Himalayan region over the period 1960–2006. Panwar et al. [16] reported a decreasing trend in pre- and post-monsoon rainfall (1969–2015) in the Shimla district of Himachal Pradesh in India, along with a significantly increasing trend in the minimum temperature. The presence of these trends suggests that there may be scarcity of water in the near future in the Himalayan region, where natural springs supply a large amount of water for irrigation. Unfortunately, the springs have been drying up due to rapid environmental degradation. In the hill town of Mussoorie in the Himalayan region, spring discharge reduced from 450 L $min^{-1}$ in 2008 to 365 L $min^{-1}$ in 2017 [17]. Similarly, in Devprayag, another town in the region, the natural springs witnessed more than a 50% decrease in discharge during 2012–2015 [18]. The scarcity of water during the lean season is a key challenge affecting agriculture in the region. Though the mid and high hills of the Indian Himalayas receive anannual rainfall of 1000–2500 mm, most of it flows down the steep slopes as runoff and does not remain available for agriculture and domestic uses. Therefore, the inhabitants of the Himalayan region have to struggle with inadequate water resources while leading to floods and food and livelihood insecurity [19]. Due to the constraint of limited water availability, farmers traditionally relied on the single cropping of cereals and minor millets during the rainy season and were forced to keep the field

fallow during the winter season. Hence, most of the cropping systems were far from their productive potential. The above-mentioned facts warrant the development of strategies to enhance the resilience of agriculture sectors in climate-sensitive areas such as the Indian Himalayan region.

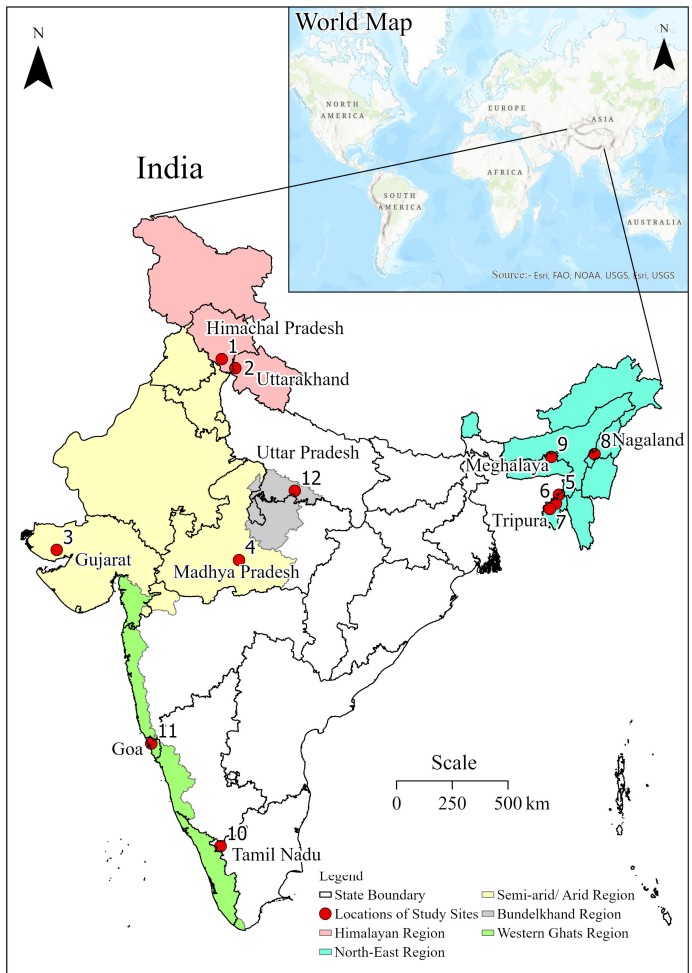

**Figure 1.** Map showing geographical location of six climatic regions and study sites.

On the contrary to the Himalayas, the semi-arid and arid regions of India (Figure 1) receive scanty annual rainfall ranging from 400–600 mm. In the semi-arid regions of India, small-sized water-harvesting ponds were successfully adopted by small landholders for sustaining agricultural production by providing supplemental or life-saving irrigation. The small-sized ponds have proven their adequacy in dealing with droughts and improving crop productivity in semi-arid lands [20]. However, such ponds are supposed to be incapable of supplying reliable irrigation supplies in the hot climates of dry arid regions [21]. In most of the semi-arid regions of India, agriculture remains mostly dependent upon groundwater resources for irrigation, which are available at relatively deeper depths and are also of poor quality due to salinity. It is true that annual rainfall in the Indian arid regions has been of small magnitude since historical times. However, it is reported that rainfall in the arid region of Kachchh district has been significantly increasing since the year 2002 under the scenario of changing climate [22,23]. It is further seen that the increase in rainfall intensity, rather than rainy days, contributed toward the rainfall increase. During the period 2007–2016, a major proportion (32–76%) of the annual rainfall was received from either 1 day of rainfall or 2–4 consecutive days of rainfall in the area [24]. The concentrated rainfall received from the short-duration storms generate huge amounts of runoff, of which a major portion quickly flows off from the catchment without providing the scope for its

utilization in agricultural production [23]. The surface water irrigation through ponds in the area is reported to have a 37.4% higher average water delivery rate than that of the tubewell supplies, and thus, the former saves about 46% of time in irrigating 1 ha of arid lands for a given irrigation depth [25]. Therefore, field-level water harvesting through farm ponds seems to be successful inenhancing agricultural production as well as water productivity in the semi-arid as well as arid regions of India.

The northeastern region of India (Figure 1) receives very high rainfall. Tripura, a constituent state of the northeastern region of India, receives an average annual rainfall of 2390 mm, of which 60% is received from June to September [26]. Orchards or commercial farms of horticultural crops do not exist in Tripura; most of the fruits and vegetables are produced at homestead plots where cultivation primarily occurs for domestic consumption. Similarly, in the adjoining state of Nagaland, the agricultural activities are largely dependent on the southwest monsoon, and the state has faced many drought or drought-like situations in the past few decades. The small and marginal farmers (85%) of the region usually concentrate on monocropping rice and mixed cropping in hilly slopes. Another state of northeastern India, Meghalaya, despite receiving the world's highest rainfall (in the Mausamgram village), still faces problems of water scarcity during post monsoon periods that ultimately affects the agricultural activities in the state. In these northeastern states, despite of the high rainfall, water scarcity exists during the post-monsoon period.

The Western Ghats mountainous region of India, covering an area of 160,000 km$^2$ (Figure 1), receives an annual rainfall ranging from 1200 to 3800 mm. Though the Western Ghats region receives relatively high rainfall, many places still experience severe water scarcity during summer months. As a result of the moisture stress and drought, the production of vegetable, fruits, and plantation crops are adversely affected.

In the Bundelkhand region of India (Central India) (Figure 1), most households live below the poverty line. The most often mentioned factor is the extreme water scarcity in the region. With the dependency of a large number of farmers on rainfed agriculture and the failure of agriculture due to recurring droughts, the level of poverty is increasing in rural areas [27,28]. Despite irrigating 45% of the net sown area in Bundelkhand, the supply of water for crop production is not satisfactory, as reported by the inter-ministerial central team [29]. The availability of groundwater is very limited in the area, as most rainwater flows out from the region without permitting its use in the agriculture sector due to rapid runoff generation in undulating topography and less permeable top soils. As a result, the region has poor groundwater potential, i.e., only 4% of the total rainfall can be stored. The promotion of the scientific planning of water conservation and rainwater harvesting activities based on the hydrological potential of watersheds is helpful for mitigating the water crisis in the Bundelkhand region [30]. The long-term rainfall data collected from 23 stations located in seven districts of the Uttar Pradesh part of the Bundelkhand region shows a declining annual rainfall (200 mm) over six decades, pushing the region into a vulnerable state [31]. The traditional system for conserving the rainwater through the earthen storage bunds of the shallow depth is prominent in the area; however, it has more evaporation due to high temperature and low humidity. Thus, it limits the availability of water to the crop after the monsoon season [32].

The situation/regions distributed throughout India elaborated above have a rainfall range of 500–11,000 mm. However, in all regions, crops face moisture stress in one season or the other, impacting crop diversity and production. In such situations, small water harvesting and recycling structures are helpful in mitigating the problems of water stress during dry spells, which further enhance the agricultural productivity. Tapping surface runoff flow in ponds is considered to be a technically feasible and economically viable option compared to tapping large dams and reservoirs. In this paper, an attempt is made to demonstrate location-specific water harvesting structures and their impact on crop production and crop diversification indices across the country.

## 2. Materials and Methods

In this study, feasibility of water harvesting under diverse climatic conditions of India is investigated by adopting a variety of methods depending upon a suitable locally available water source. Location of study sites over different states or provinces of the country is shown in Figure 1. Furthermore, the scope of utilizing the harvested water for improved crop production and diversification of cropping pattern is explored. Moreover, the possibility for scaling up location-specific water harvesting interventions at feasible locations without having any impact on the environment is worked out in different climatic regions of the country.

### 2.1. Climatic Conditions of Study Sites and Details of Water Harvesting-Cum-Irrigation Systems

Twelve case studies considered in this study are categorized into six regions based on location, climate, and terrain conditions of the individual studies. Details about the location-wise prevailing water source and water harvesting method adopted for water management and its utilization in raising agricultural and horticultural crops are provided below.

### 2.1.1. Northwestern Himalayan Region

Two locations, characterized from low to high hills, are selected, one each in the Himachal Pradesh and Uttarakhand states of the country.

(a) In Himachal Pradesh, the study area is the Kumhali village in the Shimla district, which falls under the wet temperate high hill zone (30°59′23.5″ N and 77°13′42.96″ E) and is located at an elevation of 2250 m above mean sea level (MSL). The average annual temperature is 14.6 °C. Rainfall is monsoon type, concentrated from July to September [33]. The minimum and maximum temperatures are 2 and 30 °C, respectively. The soil has a silty loam texture. The main cropping season in this relatively high-altitude region starts in March and remains until October. Thereafter, winter season descends, and temperature begins to drop and sometimes reaches to a level too low for vegetable cultivation. Vegetable production requires frequent irrigations to compensate for higher temperature and low rainfall. It is analyzed that the spring discharge may shrink and considerably vary in the area under the foreseen trends of increasing temperature and reduced rainfall under the climate change scenario. Hence, the flourishing vegetable production hub may face challenges.

Kumhali is a small village where only 14 farm families live. In the year 2015, the farm families sustained their livelihood by growing vegetables, i.e., tomato, capsicum, beans, and cauliflower, via irrigation from two natural springs. The spring discharge varied from 300 L hour$^{-1}$ (L h$^{-1}$) (May) to 2460 L h$^{-1}$ (October). The spring water is collected in a tank for nine hours by blocking the tank outlet, which is then utilized for irrigation within three hours. Thereafter, the tank outlet is blocked again, and this cycle continues. Each farm family has their turn to irrigate vegetables only a single day (three hours) in a week. The silty loam texture of soil causes high water infiltration, and hence, frequent light irrigations are warranted rather than applying infrequent heavy irrigations. Therefore, a two-way strategy is adopted to sustain the vegetable production, i.e., judicious use of water and crop diversification. In order to judiciously use the available water, the water storage of the tanks is 78.31%, and increased from 152.78 m$^3$ (in the year 2014) to 272.44 m$^3$ (in the year 2020) by constructing more tanks in other farmers' fields of the village (Figure 2). Earlier, farmers were compelled to utilize the water within three hours, leading to over-watering. However, with augmented water storage capacity, they could store more water quantities to judiciously utilize it through 2 to 3 irrigations in a week (Figure 3). Furthermore, crop diversification is achieved by encouraging farmers to grow fruit trees, such as plum, peach, apricot, and low-chilling cultivars of apple, which require less water compared to vegetables. Consequently, about 600 fruit plants were grown during 2015–2019 period (Figure 3).

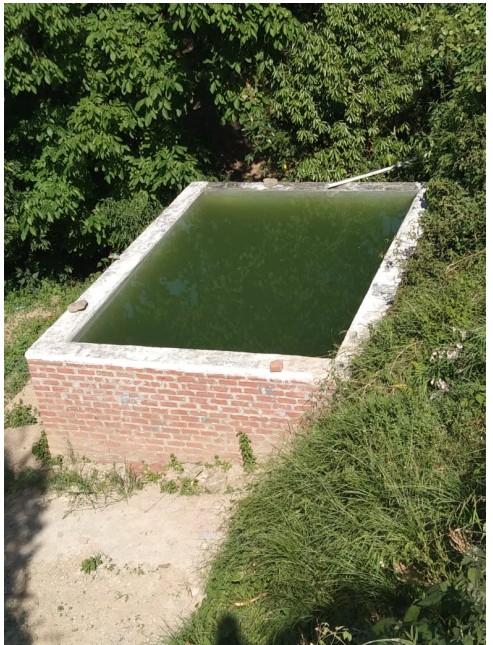
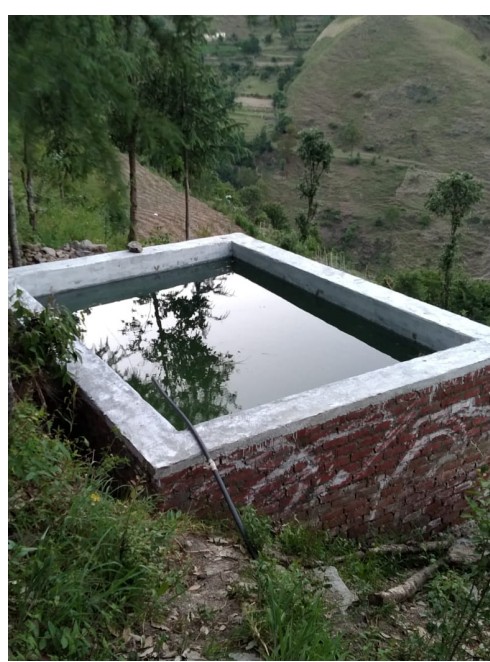

**Figure 2.** Spring-fed water storage tanks in Kumhali village of Shimla.

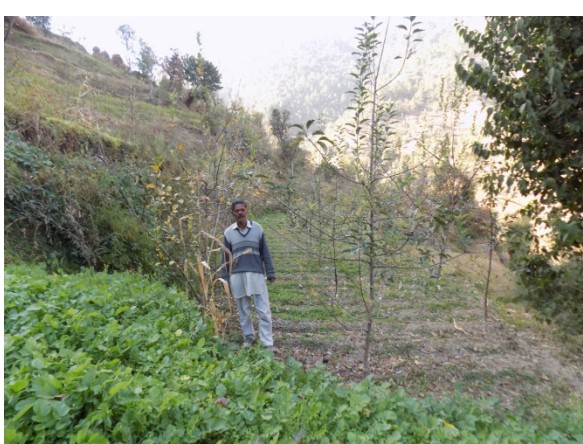
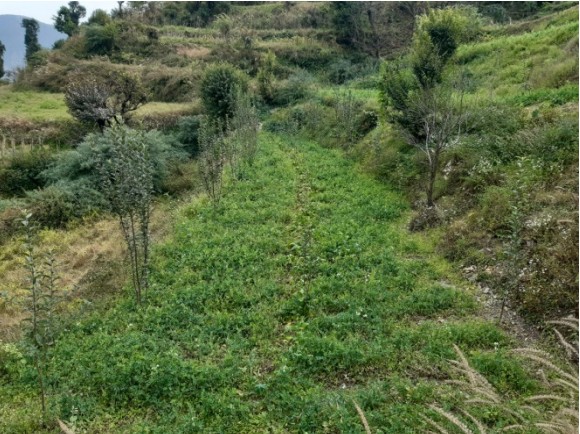

**Figure 3.** Newly introduced low-chilling cultivars of apple as crop diversification in Kumhali village.

(b) The Sahiya-Udpalta village, located at a latitude of 30°36′20″ N, longitude of 77°51′42″ E, and elevation of 1523 m MSL in the Dehradun district of Uttarakhand state, was used for the study. Climate of the area is sub-temperate, showing a prominent winter season with the average maximum and minimum temperatures of 38 and 1 °C, respectively. Rainfall is 1100 mm, the majority of which is received during monsoon season. In winter, the region receives snowfall. The soil has a silty loam texture.

Sahiya-Udpalta is a tribal village that consists of 70 households; more than half are engaged in agriculture and allied activities, i.e., animal husbandry. The village lacks assured irrigation supplies, particularly at the critical crop growth stages. In addition, the village people suffer from drinking water shortages during the lean period. The majority of the farming community belongs to the marginal and poor landholders who depend on traditional farming and/or animal husbandry activities.

A few natural springs are present in the area due to existence of mountainous terrain, which supplies water throughout the year. Looking at the perennial availability of springs' water, few springs are selected in this study for utilizing their water for vegetable production. One of the selected perennial springs, located 4 km away from the village toward

the upper reach, has a discharge of 2 L s$^{-1}$. The spring water is diverted to a chamber constructed near the spring, which later conveys the water to a tank constructed at farmer's field through a 1-inch diameter high-density polyethylene (HDPE) pipe over a distance of 4 km. Similarly, water from five other perennial springs is tapped (Figure 4). The developed water resources are utilized for raising vegetable crops, i.e., tomato, cucumber, onion, and chickpea, and to meet the domestic water requirements.

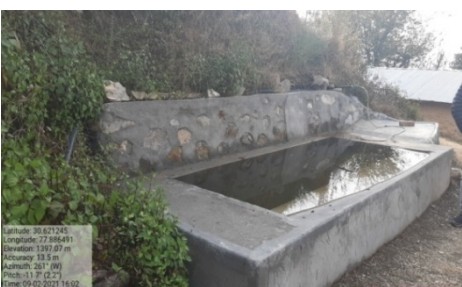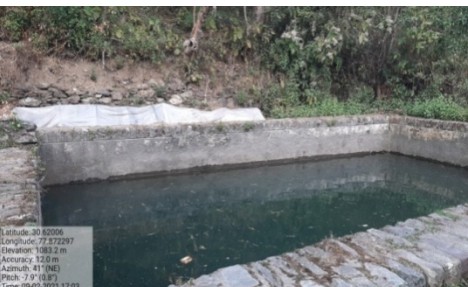

**Figure 4.** Spring water tapped, conveyed, and stored in water tank in Sahiya-Udpalta village of Dehradun.

### 2.1.2. Western Hot Arid Region

Research farm of the Regional Research Station of the ICAR-Central Arid Zone Research Institute (CAZRI) is situated at Bhuj, Gujarat, which is located at latitude of 23°13′05″ N and longitude of 69°47′20″ E, at an elevation of 120 m MSL. Climate is arid, with the minimum temperature in winter season ranging from 8.8 to 22.7 °C, and the maximum temperature ranging from 22.1 to 31.9 °C. The average annual rainfall is 389 mm. The soil is medium-textured, loamy to sandy loam, mixed hyperthermic, moderately deep lithic paleargids, developed from sandstone and shale [34].

Prior to the year 2001, the research farm of the institute lacked a secure water source for both drinking and irrigation purposes. Then, a water harvesting pond of 20,000 m$^3$ capacity (size of 180 m × 90 m and ~3 m depth) was constructed (Figure 5). Presently, the pond water is solely used to provide supplemental irrigation to winter season crops. The storage capacity of the pond was enhanced by 9184.5 m$^3$ during 2008–2009 due to high-intensity rain storms resulting in more than one filling of the pond in a year. Additionally, the capacity of the pond evidenced a reduction of 4305.5 m$^3$ over 12 years due to sedimentation [25]. The pond water is extracted from two outlets using 5 horsepower (HP) diesel-operated centrifugal pumps. The water is conveyed to the fields of wheat (1.5 ha) and mustard (2 ha) crops and applied as flood irrigation.

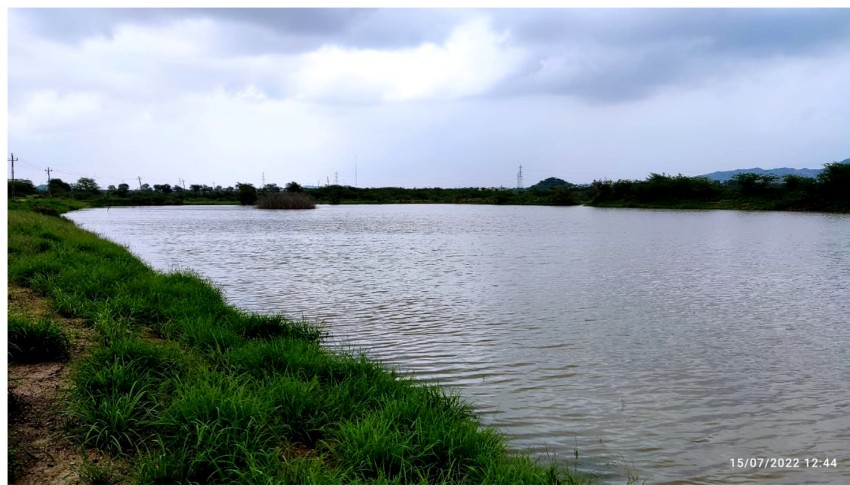

**Figure 5.** Rainwater harvesting pond constructed at Kukma village of Kachchh district.

### 2.1.3. Western Semi-Arid Region

The Navagam village in Panchmahal district in central portion of Gujarat is another site in semi-arid region, which is located at 22°49′15.10″ N latitude, 77°45′01.67″ E longitude, and 187.6 m MSL altitude. The minimum temperature ranges from 11 to 26 °C and the maximum temperature ranges from 27 to 42 °C. The average annual rainfall is 753 mm; about 90% is received during the monsoon. In the rest of the year, crops experience water stress. The soil has a silty clay texture.

In Navagam village, the majority of the farmers grow cotton, maize, pigeon pea, and horticulture crop (mango). Seeds of the high-yielding varieties of cotton (BT), maize (GM-1), pigeon pea (GT-101), and mango (Kesar) were distributed to the farmers during 2015. Two structures, i.e., jalkund (small plastic-lined water harvesting structure) (Figure 6) and recharge filter attached with dug well (Figure 7), were constructed in 2015–2016 for rainwater harvesting in order to provide need-based irrigation at the critical crop growth stages. Water storage capacity of the jalkund is kept at 4.5 m$^3$ with a 3 m length, 1.5 m width, and 1 m depth. Material of the plastic sheet is low-density polyethylene (LDPE) with a thickness of 250 microns. The dimensions of the groundwater recharge filter are 3 m in length, 3 m in width, and 1.2 m in depth. The dug well attached to recharge filter is of 4.2 m diameter and 10 m depth. One 5 HP submersible pump with a discharge of 12 L s$^{-1}$ is used for water extraction. The average recharge from the recharge filter is 3729 m$^3$. The impact assessment of the system was accomplished in the year 2022.

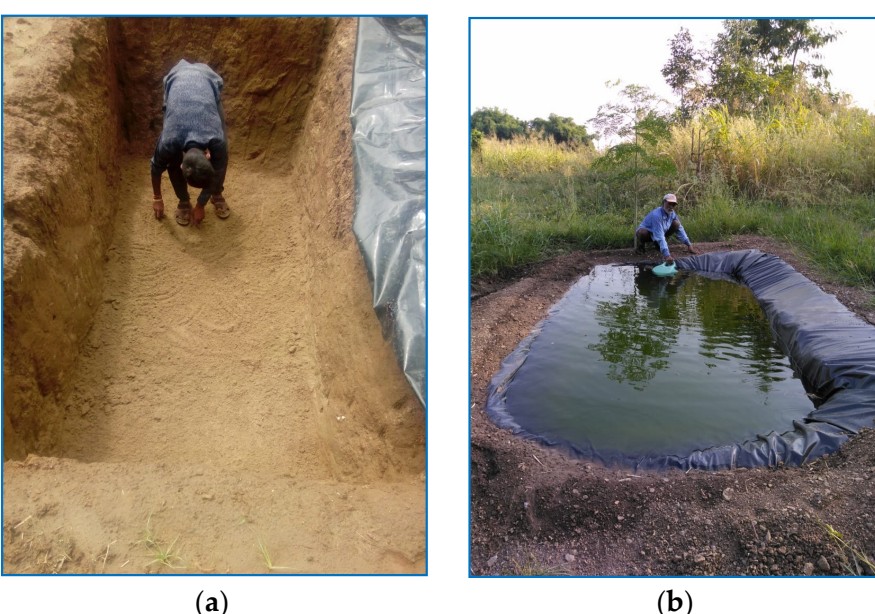

(**a**)          (**b**)

**Figure 6.** (**a**) Digging of pit in 2015 for construction of water harvesting pond (jalkund) and (**b**) plastic-lined pond.

### 2.1.4. Humid Region

Three states, i.e., Tripura, Nagaland, and Meghalaya of the humid areas of northeastern parts of India were considered for this study.

(a) In humid regions, the agricultural strategies evolved and/or adopted by farmers to deal with the water-related problems under three different scenarios, i.e., water logging during monsoon (July to September), water scarcity during non-monsoon (October to June), and unpredictable water supply, are understood through questionnaire-based interviews. Furthermore, studies on water management and utilization in agricultural production are undertaken in three villages of Tripura.

(i) The Kacharghat village in Kailashahar subdivision of Unakoti district is located between 24°19′40.85″ N latitude and 92°00′08.77″ E longitude with an altitude of 26 m MSL. The minimum temperature in the winter season ranges from 10.0 to 28.0 °C, and

the maximum temperature ranges from 25.25 to 33.7 °C. The average annual rainfall is 381.84 cm. The soil has a sandy loam to clay loam texture.

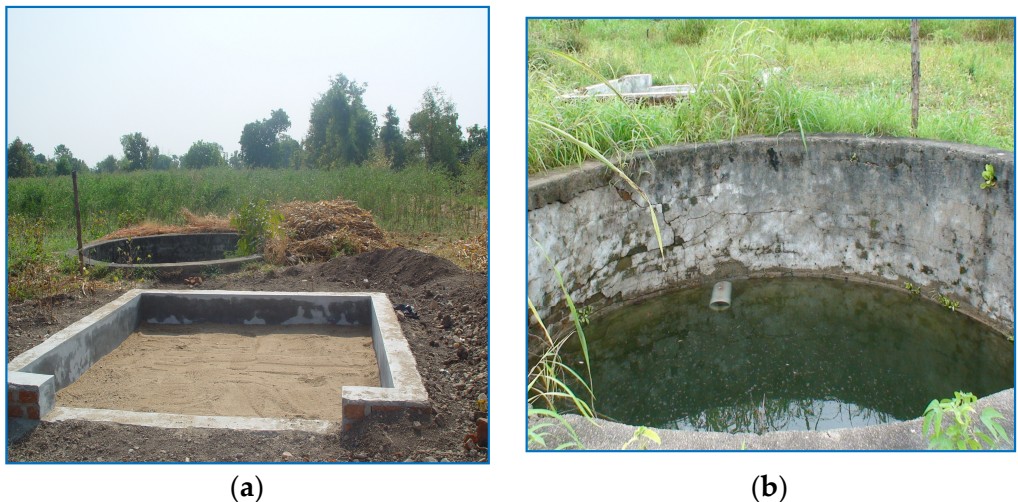

(**a**)　　　　　　　　　　　　　　　　(**b**)

**Figure 7.** (**a**) Recharge filters attached with dug well, and (**b**) dug well at its full water storage capacity.

In Kacharghat village, 1.69 ha of land suffered from water logging, rendering the land unusable for agriculture and residence. In the years 2006–2007, a culvert was constructed for safe disposal of excess water from the waterlogged land. Hence, a sluice gate was further established in the year 2012 at the mouth of the culvert to regulate the water flow through the waterlogged field after retaining 1 m water depth. With this intervention, 1.69 ha of fallow waterlogged land (locally called doba) was converted to an artificial wetland where cultivation of *Colocasia esculenta* (taro), fish, and edible aquatic plants of *Enhydra fluctuans* was undertaken (Figure 8).

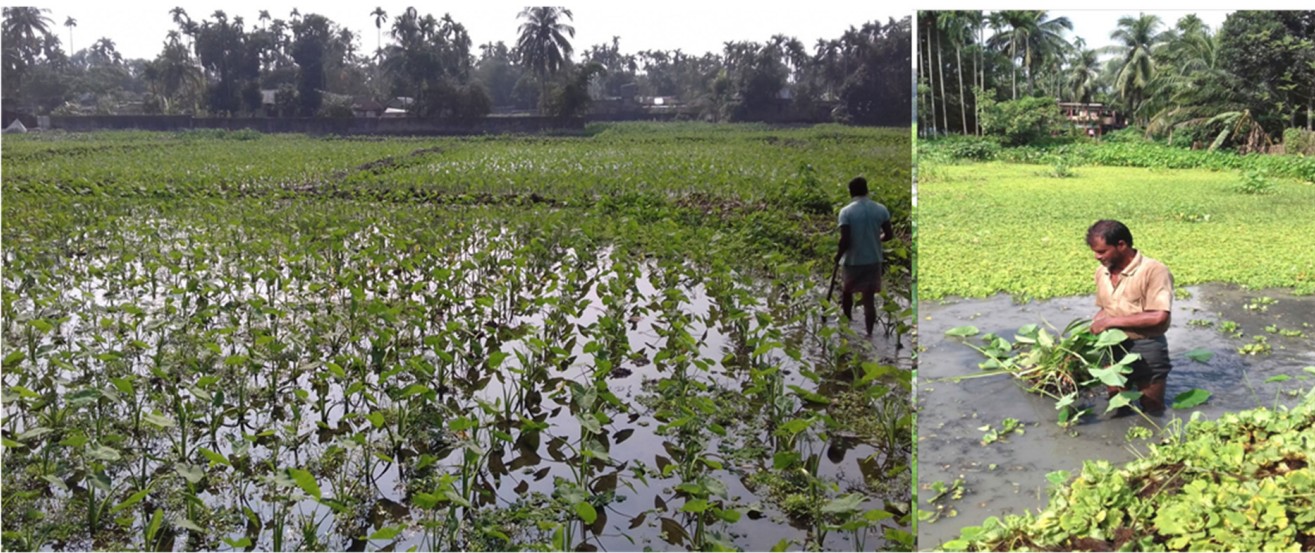

**Figure 8.** Waterlogged land converted into artificial wetland for cultivation of aquatic plants in Kacharghat village of Tripura.

(ii) The Purba Daluchhara village in Salema subdivision of Dhalai district is located at 23°59′56.83″ N latitude, 91°30′24.46″ E longitude, and 125 m MSL altitude. The minimum temperature in the winter season ranges from 7.0 to 28.0 °C, and the maximum temperature

ranges from 28.0 to 34.5 °C. The average annual rainfall is 381.84 cm, and the soil has a sandy clay loam texture. It suffers from water scarcity for paddy cultivation.

In Purba Daluchhara village, a farm family owned 0.32 ha land, where they used to grow paddy and vegetables in rotation. Initially, irrigations depended on the water allotted from the Daluchhara stream at weekly intervals for the specific time periods as decided by the village administration. However, the sole dependence of irrigation on the stream-fed water hampered the paddy production. Hence, in the year 2010, 0.16 ha area under paddy cultivation was spared for construction of a multipurpose earthen pond with 1.5 m depth to serve both fish as well as crop cultivation (Figure 9). The pond is managed along the lines of integrated multi-trophic aquaculture strategy by involving aquatic edible plants (*Enhydra fluctuans*, *Ipomea aquatica*), freshwater mussels (*Lamellidens* spp.), and fish (Indian major and medium carp) as components.

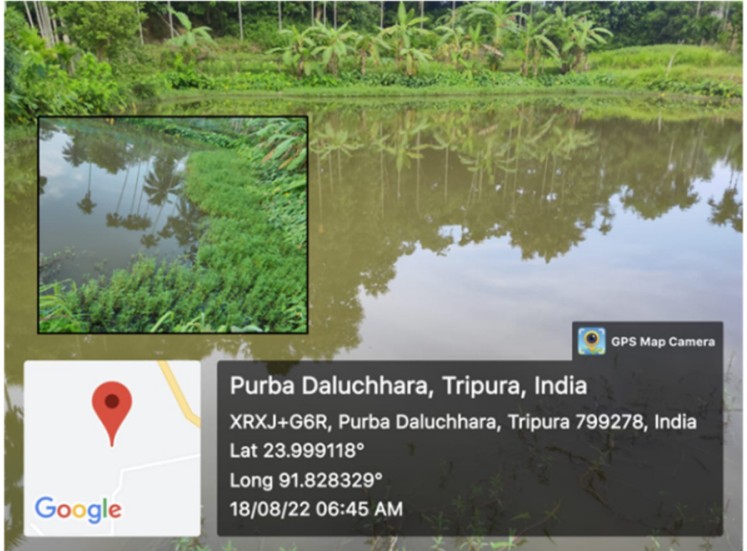

**Figure 9.** Multipurpose earthen pond constructed in Purba Daluchhara village of Tripura to store stream-fed water for aquaculture and irrigation to paddy and vegetables.

(iii) The Hathai Kotor village in Jirania subdivision of West Tripura district is located at 23°59′56.83″ N latitude and 91°30′24.46″ E longitude with altitude of 82 m MSL. The minimum temperature in the winter season ranges from 12.2 to 25.2 °C, and the maximum temperature ranges from 24.6 to 32.1 °C. The average annual rainfall is 200.39 cm, and the soil is red laterite that suffers from water scarcity in the hilly part of the landscape.

In Hathai Kotor village with undulating topographic terrain, highlands (locally called *tilla*) are used for residential purposes, while the low-lying plain areas are used for paddy and fish cultivation using water supplied from the constructed ponds. Apart from the rains, the major source of water for cultivation of paddy and fish is the Howrah River. The water ponds are mainly located in the low-lying areas, and thus, these are unable to supply water for irrigation and domestic needs of homestead plots located in the highlands. Consequently, a large proportion of area in highlands remains uncropped or under-cropped. To circumvent this issue, plastic-lined rainfed ponds (jalkunds) with capacities ranging from 10,000 to 30,000 L were introduced in the highlands for irrigation (Figure 10). The ponds were constructed at the highest altitude in farmers' fields under shaded or sheltered pockets to minimize evaporation loss. The natural gradient of the pond sites was utilized for irrigating horticultural crops in the middle altitudes using the gravity flow of water, thus avoiding the need for energy for water conveyance.

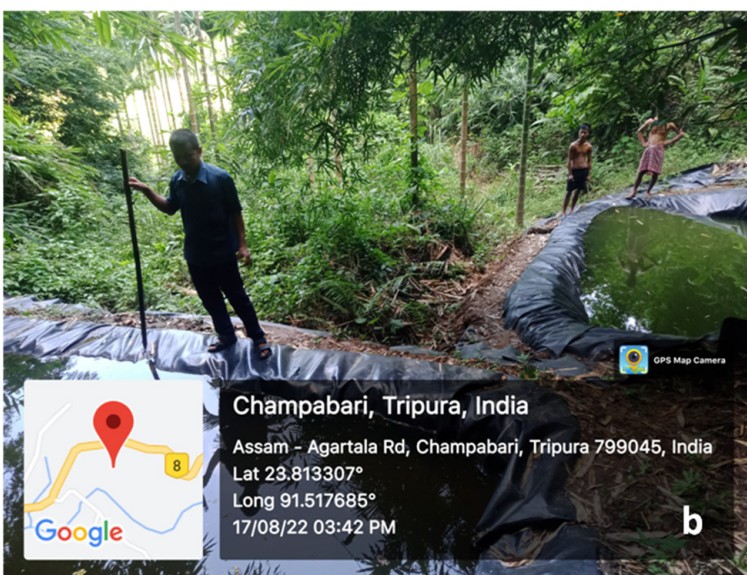

**Figure 10.** Plastic-lined water storage pond (jalkund) fed from Howrah River flowing through Hathai Kotor village of Tripura.

(b) The New Medziphema village in Chümoukedima district of Nagaland is located at an elevation of 320 m MSL, between 25°45′15.03″ N latitude and 93°50′39.99″ E longitude, with an average annual rainfall of 1558 mm, where the maximum precipitation occurs during the monsoon. The average monthly maximum and minimum temperatures vary from 22.4 to 31.4 °C and from 9.4 to 25.2 °C, respectively (source: average of 1999–2020 data, Agromet Observatory and Automatic Weather Station, ICAR Nagaland Centre, Medziphema). Soils of the region are sandy loam and acidic in reaction.

In New Medziphema village, rainwater is harvested by constructing microstructures (jalkund) of 5 m × 4 m × 1.5 m in size (Figure 11). The harvested water is subsequently utilized for irrigating winter season vegetables, i.e., cabbage, French bean, brinjal, and broccoli, through drip and sprinkler. Rice is the major crop in the area, followed by maize, soybean, and some vegetables (very negligible area) grown during the *kharif* or rainy season. Additionally, farmers started planting some fruit crops such as pineapple, banana, etc.

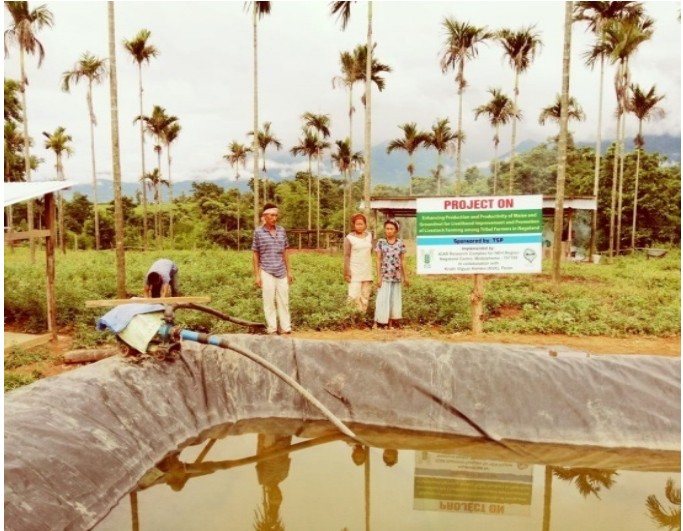

**Figure 11.** A small-sized plastic-lined pond (jalkund) for rainwater harvesting at a farmer's homestead plot in New Medziphema village of Nagaland.

(c) The Borgang village in the Marngar cluster of the Umling block is located at an elevation of 608 m MSL in Ri-Bhoi district of Meghalaya. The longitude and latitude of the area are 25°53′32.96″ N and 91°54′54.49″ E, respectively. On average, the months of April and December registered the hottest (29.4 °C) and the coldest (7.6 °C) temperatures in a year. This region receives a mean annual rainfall of 1637 mm, where most of the rainfall occurs during the monsoon. The soil of this region has a loamy to fine loamy texture.

In the Borgang village, a low-cost earthen pond of 60 m × 30 m × 2 m in size is constructed to sustain the vegetable production during winter season (Figure 12). Earlier, only rainfed vegetables such as squash, turmeric, and pumpkin could be grown. However, after construction of the pond, crop diversification was possible due to availability of irrigation facilities. Consequently, commercial vegetables such as broccoli, cauliflower, cabbage, and radish are successfully introduced during the winter season. Apart from the vegetables, banana is also planted on the boundary of the pond (Figure 13).

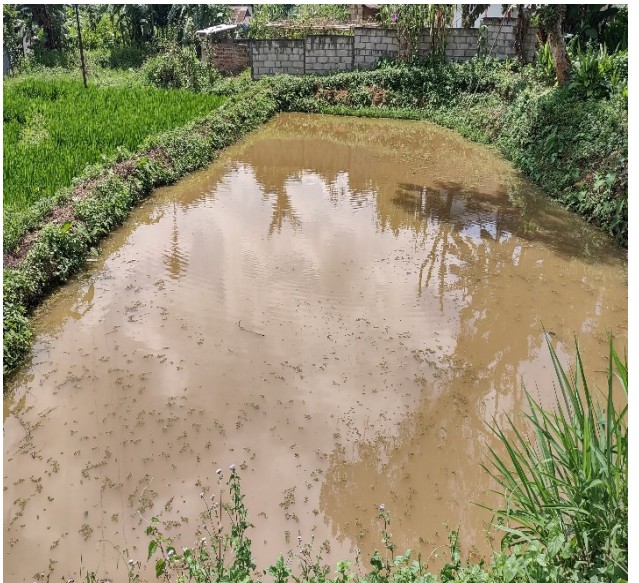 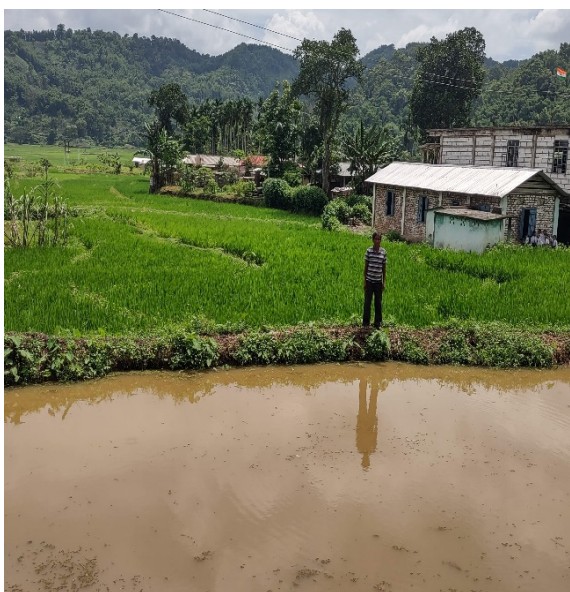

**Figure 12.** A low-cost earthen pond constructed for rainwater harvesting in Borgang village of Meghalaya.

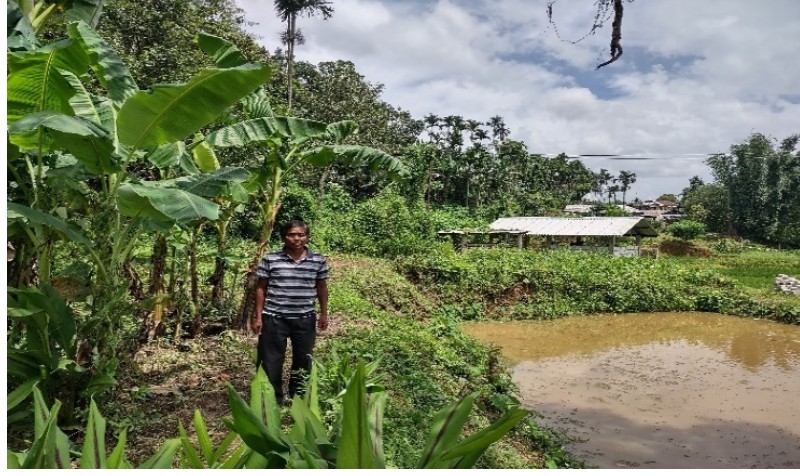

**Figure 13.** View of horticultural crops produced from pond water irrigation in Borgang village.

### 2.1.5. Western Ghats Mountainous Region

The Western Ghats of India stretches from Goa to Tamil Nadu, and hence, two locations representing the situation in the two states are undertaken in this study.

(a) The first study was conducted at the research farm of the ICAR-Indian Institute of Soil and Water Conservation in Ooty, Udhagamandalam, Tamil Nadu. The study area is located at 11°23′56.23″ N latitude, 76°40′08.96″ E longitude, and 2183 m MSL altitude. The average annual rainfall is 1325 mm, which is received through southwest and northeast monsoons. About 55% of the rainfall is received during the monsoon. The average annual temperature varies from 10 to 25°C. The major crops of the study area are vegetables, and the soil has a sandy clay loam texture.

An unlined farm pond with a storage capacity of 2800 m$^3$ (45 m × 25 m × 2.5 m) was constructed in the year 2017 at the research farm of the institute. The stored pond water is used to provide supplemental irrigation to vegetables, namely, beans and potato, grown in 0.5 ha during a period of scarce water availability (Figure 14). Daily inflow and outflow data of the pond, i.e., water level and evaporation, were monitored for three years, i.e., 2018–2021. The average water availability for irrigation was worked out to be 1850 m$^3$ based on water balance method.

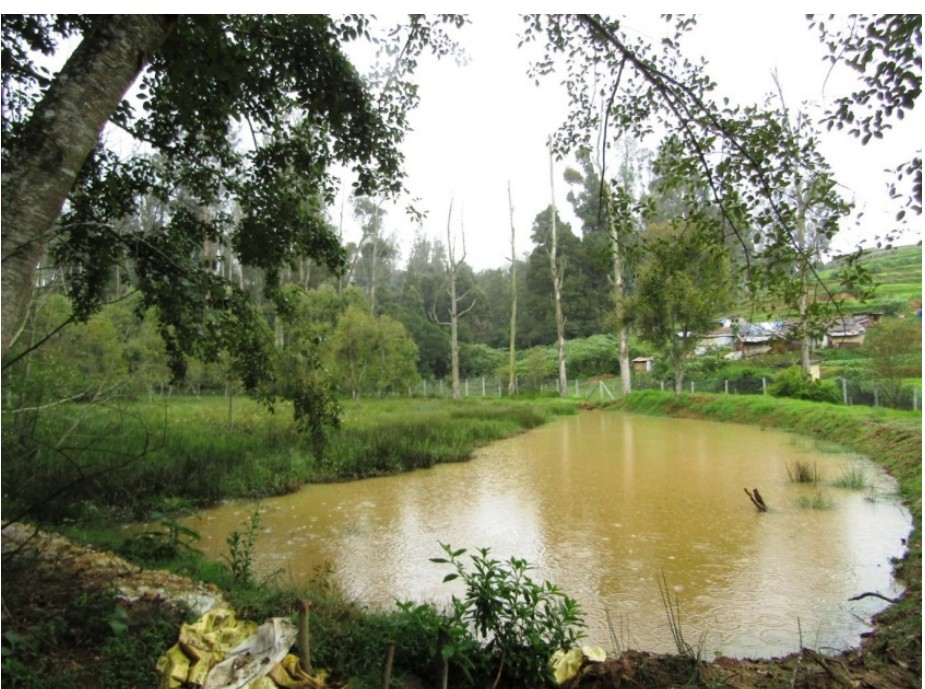

**Figure 14.** Unlined farm pond at full storage capacity at Ooty in Tamil Nadu.

(b) The second study is conducted at ICAR-Central Coastal Agricultural Research Institute, Ela, Old Goa, located at 15°29′28″ N latitude, 73°55′14″ E longitude, and 67 m MSL altitude. The average annual rainfall is 2892 mm, which is received through southwest monsoon. About 80% of the rainfall is received during the monsoon. The average annual temperature varies from 28 to 36 °C. The major crops are mango, cashew, arecanut, and coconut. The soil of the study site is sandy loam in texture.

In old Goa, a plastic-lined farm pond of 50 m in length, 25 m in width, and 2.8 m in depth with a storage capacity of 3500 m$^3$ was constructed to harvest rainwater and supply supplemental irrigation to plantation crops during water scarcity period (Figure 15). A silpauline polyfilm of 300 GSM (g m$^{-2}$) is used for pond lining in order to check the seepage loss. The harvested water is used for irrigating cashew (0.40 ha) and mango (0.60 ha).

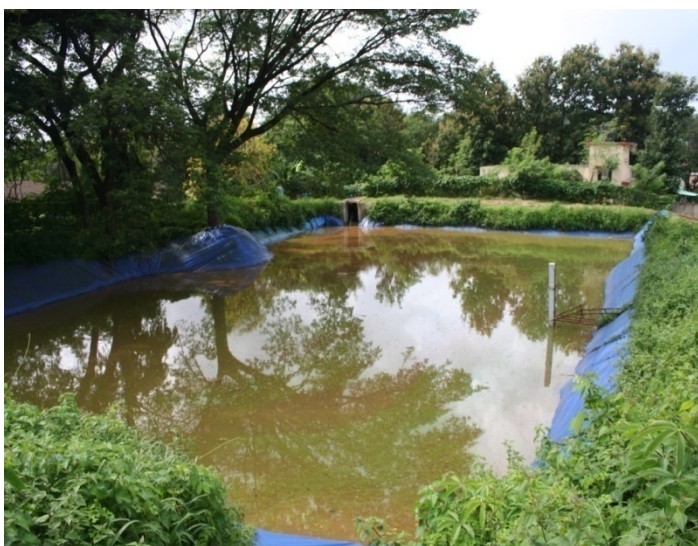

**Figure 15.** Lined farm pond at full storage capacity at Old Goa, Goa.

At both sites in the Western Ghats mountainous region, the harvested rainwater is utilized for providing supplemental irrigation during dry spells in monsoon and/or for sustaining additional crop or intercrop during *rabi* or winter season. An increase in productivity and income is monitored with and without farm pond situations.

2.1.6. Central or Bundelkhand Region

In the Chahitara village of Banda district, Uttar Pradesh, the research farm of Banda University of Agriculture and Technology (BUAT) is located at 25°31′47.12″ N latitude and 80°19′59.62″ E longitude, and at an elevation of 122 m MSL. The climate is semi-arid, with the mean monthly minimum and maximum temperatures of 5.8 and 47 °C, respectively. This region receives a mean annual rainfall of 902 mm, with a large fraction of the rainfall received during the monsoon season. The soil of this region is mostly silty clay in texture.

In the study area, four farm ponds of the dugout-cum-embankment type were constructed in the year 2016 (Figure 16). The storage capacity of the 1st, 2nd, 3rd, and 4th ponds is 10,500, 7040, 20,000, and 24,000 m$^3$, respectively, with a total of 61,540 m$^3$. Water stored in the ponds has been used for irrigating winter season crops as well as fulfilling domestic water requirements of 100–150 cattle since 2017–2018. The pond water is extracted using centrifugal pumps operated through solar (1st and 2nd ponds), electric (7.5 HP motor at 1st pond), and diesel engine (8 HP motor at 2nd pond) power. The extracted water is carried up to crop fields using HDPE section pipes of 4-inch diameter. At the 3rd and 4th ponds, only diesel engine-operated pump is used for water extraction due to non-availability of electricity. The water stored inside the 1st and 2nd ponds is fully utilized compared to the 3rd and 4th ponds, as most of the cultivable land is situated nearby the 1st and 2nd ponds.

*2.2. Recording Observations and Data Analysis*
2.2.1. Crop Production and Diversification Indices

In almost all the case studies, the data related to the cropped area and production are recorded for all the crops, which are grown through irrigation provided from the water harvesting or storage system. In addition, the data of the average consumption as well as the recommended dose of applied fertilizers in terms of nitrogen (N), phosphorus (P), and potassium (K) are also recorded at all the sites except for the Sahiya-Udpalta village of Dehradun, the Kukma village of Kachchh in Gujarat, and the BUAT campus of Banda in Uttar Pradesh. It is worth mentioning that all inputs such as land preparation, intercultural operations, fertilizers, pesticides, etc., are kept similar and as per standard practice adopted in the region before and after the implementation of water harvesting interventions at all individual sites. Thus, except for water application, the impact of all

other parameters is the same at each individual site for before and after situations of the water harvesting interventions. Water stored inside the dedicated structures is monitored at every site. Proportion of total irrigated crop area under each individual crop is computed at nine sites. Yield of all crops is calculated based on production and cropped area at all 12 sites. Production under each crop per unit of applied water is estimated for nine sites. The observations recorded at nine sites are used to compute gross productivity and gross returns, in addition to six indices related to crop production and diversification, i.e., cropping intensity (CI) at a single site, cultivated land utilization index (CLUI), crop productivity index (CPI), crop diversification index (CDI), crop water productivity index (CWPI), and crop fertilization index (CFI). Adequacy of water bodies in storing water is tested by estimating storage efficiency (SE). Details of six evaluation criteria used for assessing comparative performance of the water storage structures in improving crop yield and production are summarized in Table 1.

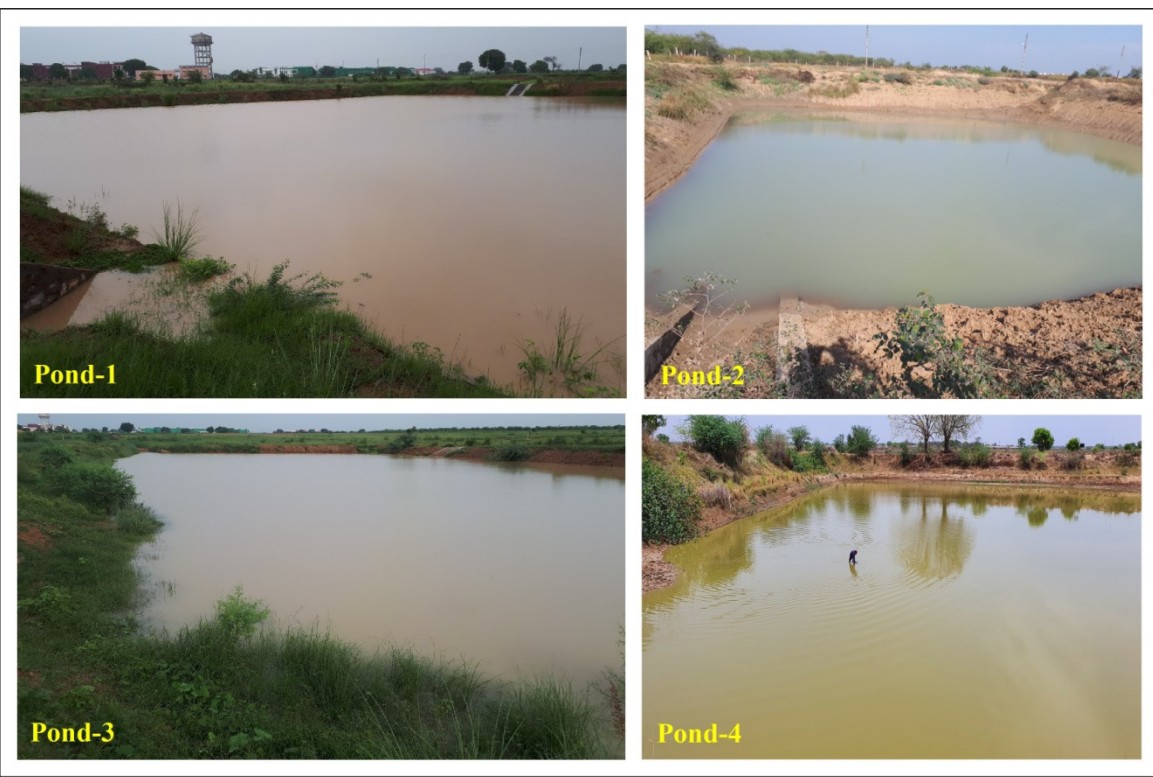

**Figure 16.** Four ponds at full storage capacity at BUAT Campus, Banda.

### 2.2.2. Economic Evaluation Criteria

Gross returns obtained from the sales of the crops are estimated based on the crop yields, their acreages, and the prevailing market rates at nine sites. At three sites, i.e., Sahiya-Udpalta village of Dehradun in northwestern Himalayan region, Kukma village of Gujarat in hot arid region, and Chahitara of Uttar Pradesh in central region, the cost of pond construction as well as the cost of cultivation for crops involving all kinds of inputs are considered for calculating the economic returns. Net returns are computed by subtracting total cost from gross returns. Economic evaluation of the pond is performed using three indicators, i.e., benefit–cost ratio (BCR), net present value (NPV), and internal rate of return (IRR). Details of these three economic evaluation criteria are summarized in Table 1. At Kukma (Gujarat) site, the unit cost of harvested rainwater is computed under both the best-case (pond at full storage capacity every year) and worst-case (pond at half of the full storage capacity) scenarios.

**Table 1.** Summary of crop production and diversification indices used to evaluate performance of water harvesting interventions in this study.

| S. No. | Name of Index/Indicator | Expression | Reference |
|---|---|---|---|
| 1 | Cropping Intensity (CI) | $\text{CI} = \dfrac{\text{Gross cropped area}}{\text{Net sown area}}$ | |
| 2 | Cultivated Land Utilization Index (CLUI) | $\text{CLUI} = \dfrac{\sum_{i=1}^{n} a_i d_i}{A \times 365}$ | |
| 3 | Crop Productivity Index (CPI) | $\text{CPI} = \dfrac{1}{n} \sum_{i=1}^{n} (y_i / Y_i)$ | |
| 4 | Crop Diversification Index (CDI) | $\text{CDI} = \sum_{i=1}^{n} P_i \log(1/P_i)$ | Sharda et al. [35] GoI [36] |
| 5 | Conserved Water Productivity Index (CWPI) | $\text{CWPI} = \dfrac{\sum_{i=1}^{n} Y_{ai}}{\sum_{i=1}^{n} Y_{ti}}$ | |
| 6 | Crop Fertilization Index (CFI) | $\text{CFI} = \dfrac{\sum_{j=1}^{f} (NPK)_{cj}}{\sum_{j=1}^{f} (NPK)_{rj}}$ | |
| 7 | Storage Efficiency (SE) | $\text{SE} = \dfrac{\sum_{k=1}^{w} Q_{si}}{\sum_{k=1}^{w} Q_{di}}$ | |
| 8 | Benefit–Cost Ratio (BCR) | $\text{BCR} = \sum_{t=1}^{y} \left[ \dfrac{B_t}{(1+r)^t} \right] / \sum_{t=1}^{y} \left[ \dfrac{C_t}{(1+r)^t} \right]$ | Brooks et al. [37] |
| 9 | Net Present Value (NPV) | $\text{NPV} = \sum_{t=1}^{y} \left[ \dfrac{B_t - C_t}{(1+r)^t} \right]$ | |
| 10 | Internal Rate of Return (IRR) | $\sum_{t=1}^{y} \dfrac{B_t}{(1+IRR)^t} - \sum_{t=1}^{y} \dfrac{C_t}{(1+IRR)^t} = 0$ | Yuan et al. [38] |

Note: $a_i$ = area occupied by ith crop; $d_i$ = days when ith crop occupied $a_i$ area; A = cultivated land area; n = number of crops; $y_i$ = average yield of ith crop cultivated in targeted area; $Y_i$ = yield of ith crop with standard package of practices; $Y_{ai}$ = average production of ith crop achieved (equivalent yield) per unit water; $Y_{ti}$ = targeted production of ith crop (equivalent yield) per unit water; $(NPK)_{cj}$ = average consumption of jth fertilizer in terms of N, P, and K; $(NPK)_{rj}$ = recommended or required dose of jth fertilizer in terms of N, P, and K; f = number of fertilizers; $Q_{si}$ = actual water stored in kth water body; $Q_{di}$ = designed live storage capacity of kth water body; w = number of water bodies; $B_t$ = benefit in tth year; $C_t$ = cost in tth year; r = discount rate; y = number of years.

### 2.2.3. Evaluating Significance of Water Harvesting Intervention

The values of change in crop production and diversification indices indicate the impact of water harvesting intervention on agricultural production. Hence, these indices are subjected to paired-sample Wilcoxon signed-rank test to assess the statistical significance of the change in their values before and after the interventions. The Wilcoxon signed-rank test, a nonparametric statistical test, is performed by utilizing a few salient aspects of sample data, i.e., signs of measurements, order relationships, or category frequencies. Thus, the test does not require any underlying assumption such as normality in the population data unlike the parametric tests, and hence, it is distribution-free. The Wilcoxon signed-rank test is a commonly used nonparametric test for paired data, e.g., consisting of before and after measurements [39]. In this study, the test is applied using R software, version R.4.2.2 [40]. The null hypothesis ($H_0$) of the test is considered as no difference in value of the indices before and after the intervention, which indicates that the water harvesting intervention does not have any impact on agricultural production. On the contrary, the alternative hypothesis ($H_1$) is that the water harvesting intervention has a significant impact on agricultural production.

For the paired data of n samples, $(x_1, y_1), \ldots, (x_n, y_n)$, differences between all paired values, i.e., $d_i = x_i - y_i$, are computed, where $x_i$ and $y_i$ are the ith responses before and after the intervention, respectively. Then, ascending ranks ($r_i$) are assigned to all the $d_i$s in increasing order from the smallest to the largest absolute values. Ties are assigned average ranks, and thereafter, signs + or − are attached to the ranks ($r_i$) according to the positive or negative $d_i$s values, i.e., $\text{sgn}(d_i)$. Sums of all positive and negative ranks are separately

computed. Finally, the test statistic (T) is defined as the smaller of the two absolute sums of positive ($+r_i$) and negative ($-r_i$) ranks, and is given as follows [41]:

$$T = \text{Minimum} \left( \sum |+r_i|, \sum |-r_i| \right) \tag{1}$$

The value of T along with the sample size n are used to find the *p*-value from the standard table of the Wilcoxon signed-rank test. The $H_0$ is rejected if the *p*-value < 0.05, which indicates that the water harvesting intervention has a significant impact on the change in crop production and diversification indices at a 5% significance level.

### 2.3. Scope of Extending Water Harvesting Interventions at Large Scale

This study explores the potential of further expanding the domain of location-specific water harvesting interventions, focused on the presented case studies, in different climatic regions of the country. While discussing the scaling-up potential of the water harvesting interventions, only feasible areas are considered where the said intervention may be implemented.

### 3. Results

#### 3.1. Impact of Water Harvesting on Crop Production

The values of six crop production and diversification-related indices along with the gross productivity and gross returns both before and after water harvesting/storage intervention for nine sites are presented in Figure 17. The detailed results for the different climatic regions are inferred ahead (Figure 17).

#### 3.1.1. Northwestern Himalayan Region

In the Kumhali village, the impacts of water harvesting activities are assessed for the year 2022. It is observed that the cultivation land utilization index (CLUI) increased by 38.5% as a result of the increased water storage. This finding signifies that arable lands are better utilized in terms of cropped area and/or crop duration with the augmented availability of irrigation water. The crop productivity index (CPI) shows a slight increase of 4% after the water storage intervention. It is further seen that the crop diversification index (CDI) after the intervention increased by 14% due to the introduction of fruit crops. However, the value of the crop fertilization index (CFI) after the intervention (1.05) is closer to 1.0 with a decrease of 11% compared to before the intervention (1.18). The involvement of fruit crops implies that the risk of monetary losses due to crop failure is minimized. Moreover, the gross productivity and gross returns indicate a remarkable growth of 28 and 20%, respectively, after the intervention of water storage.

In the Sahiya-Udpalta village, the farmers realized more than a 2.5-fold increase in the unit area net income (INR 133,340 ha$^{-1}$) after the intervention of water harvesting/storage structure compared to the traditional agriculture system (INR 52,435 ha$^{-1}$) due to the availability of irrigation water and the introduction of high-yielding vegetable varieties (Table 2).

#### 3.1.2. Western Hot Arid Region

In the Kukma village, it was realized over a period of 10–12 years that the rainwater fills the pond to its full storage capacity, receiving about 250 to 300 mm of annual rainfall in the area. It was further observed that the pond has at least one filling in most of the years since 2001. This finding emphasizes the scope of rainwater harvesting in the arid region of northern and northwest Gujarat. This became possible mainly due to the occurrence of intense rainy storms in the area under the climate change scenario. The unit cost of harvesting rainwater in arid land comes to be INR 1.51 m$^{-3}$ for the 30-year life of the pond, which is considerably low, and thus, reveals that rainwater harvesting in the Indian arid region is now becoming economically viable. The estimates of the benefit–cost ratio (BCR) (1.01), net present value (NPV) (INR 10,093), and internal rate of return (IRR) (10%) for a 30-year period as the productive life of the pond further proves the efficacy of the

rainwater harvesting pond in economically supplying supplemental irrigation to arid crops (Table 3). The calculation of the economic indicators, i.e., BCR, NPV, and IRR, involves the computation of incremental cost as well as return in each individual year over a 30-year period, which is later on converted to the present worth of the total cost and return. Moreover, the adoption of the water-efficient irrigation practice may further lead to the expansion of irrigated cropped area that will lead to enhanced food security and added benefits. Under the water-efficient irrigation practice, the results reveal the possibility of enhancing irrigated areas by 73 and 70% under wheat and mustard crops, respectively, with a 71% overall increase in cropped areas. In addition, the water-efficient irrigation system may result in better values of the economic indicators, with an increase of 2.18 times in the BCR, an increase of more than 100 times in the NPV, and an increase of 2.39 times in the IRR over the current practice (Table 3).

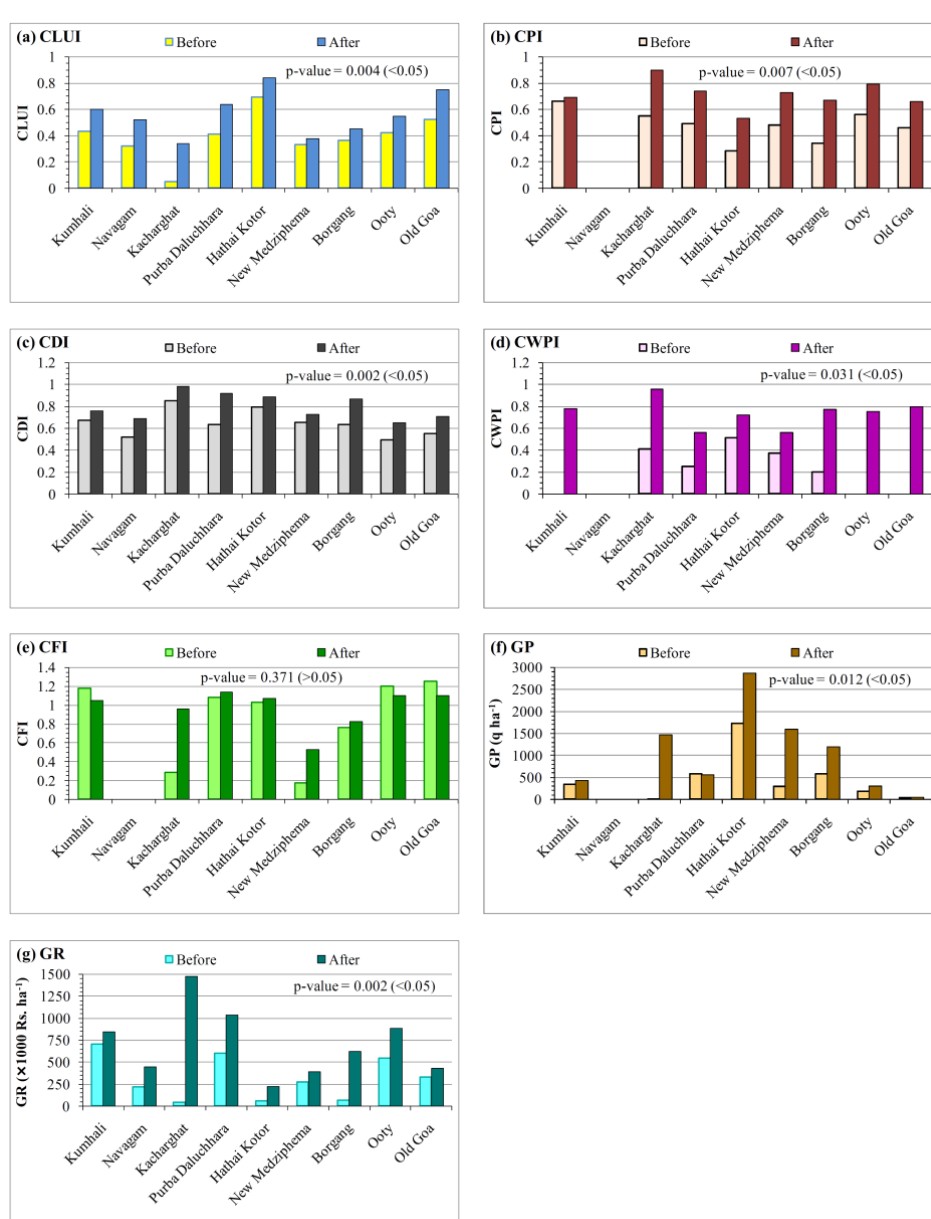

**Figure 17.** Bar charts showing values of crop production and diversification indices before and after implementing water harvesting interventions at 8 sites. (**a**) CLUI—cultivated land utilization index; (**b**) CPI—crop productivity index; (**c**) CDI—crop diversification index; (**d**) CWPI—crop water productivity index; (**e**) CFI—crop fertilization index; (**f**) GP—gross productivity; (**g**) GR—gross returns.

**Table 2.** Production of high-yielding crops under diversified agriculture systems (Sahiya-Udpalta village, Dehradun).

| Name of Crop Component | Area (ha) | Production (tha$^{-1}$) | Gross Income (INRha$^{-1}$) | Net Income (INRha$^{-1}$) |
|---|---|---|---|---|
| Tomato (Cv. Himsona) | 0.24 | 4.5 | 74,250 | 57,375 |
| Cucumber (Cv. Malini) | 0.12 | 2.26 | 27,180 | 14,242 |
| Onion (Cv. Agri-Found Light Red) | 0.12 | 3.30 | 26,400 | 18,460 |
| Green Pea (Cv. Golden GS-10) | 0.20 | 2.26 | 57,051 | 36,433 |
| Paddy | 0.16 | 0.75 | 6750 | 3730 |
| Toria (Cv. Hill-1) | 0.20 | 0.11 | 8100 | 3100 |
| Fellow Land (Fruit Trees) | 0.16 | At present in juvenile phase | - | - |
| Total | 1.20 | 13.19 | 199,731 | 133,340 |

**Table 3.** Economic analysis of farm pond under existing practice of over-irrigation and water-efficient irrigation system (Kukma village of Kachchh).

| Item | Existing Practice of Over-Irrigation | Water-Efficient Irrigation System |
|---|---|---|
| Present worth of incremental return(INR) | 1,140,855 | 1,130,762 |
| Present worth of total cost @ 10% (INR) | 2,461,320 | 1,130,762 |
| Benefit–cost ratio | 1.01 | 2.18 |
| Net present value (INR) | 10,093 | 1,330,558 |
| Internal rate of return (%) | 10.12 | 24.23 |

Note: INR = Indian rupees (INR 79.96 = USD 1) conversion rate on 18 July 2022. Source: https://www.xe.com/currencyconverter/convert/?Amount=1&From=USD&To=INR (accessed on 18 July 2022).

3.1.3. Western Semi-Arid Region

In the Navagam village, high-yielding varieties and irrigation from water harvesting structures (WHS) significantly improve the yield of food and fruit crops grown in the study area. The average of the data over six years (2016–2021) reveals that the crop yield of cotton, maize, and pigeon pea increased by 25.4, 23.2, and 20.3%, respectively, compared to their local cultivars grown under limited irrigation (Table 4). Further, the newly introduced horticulture crop, i.e., mango, flourishes well at the site, and its economical yield starts after three years of plantation. The 3-year average fruit yield of the mango plantation is about 4.5 t ha$^{-1}$. Similarly, the cropping intensity doubled from 200 to 400% after the water storage intervention, and the value of the CLUI increased by 62.5%. The water use efficiency (WUE) of cotton, maize, and pigeon pea shows an increase of 20, 14, and 23.8%, respectively, compared to their local cultivars (Table 4). The storage efficiency of jalkund and the dug well is observed to be 100% during the peak rainfall period. In addition, the adoption of the high-density crop planting method and the introduction of fruit crops improve the value of CDI by 17.3%. This finding suggests that the crop diversification after the implementation of jalkund and the dug well minimizes the risk of crop failure under stress conditions as the fruit crops, which are harder than the food crops, sustain growth and yield fruits even under less water availability.

**Table 4.** Impact assessment of water harvesting structure after six years of implementation (Navagam village of Gujarat).

| S. No. | Parameter | Value | |
|--------|-----------|-------|---|
| | | Before Project | After Project |
| 1 | Crop yield (t ha$^{-1}$) | | |
| | Cotton | 1.0 | 1.25 |
| | Maize | 2.5 | 3.08 |
| | Pigeon pea | 1.2 | 1.44 |
| | Mango | – | 4.5 |
| 2 | Cropping intensity (%) | 200 | 400 |
| 3 | Water use efficiency (kg ha$^{-1}$ mm) | | |
| | Cotton | 5.0 | 5.7 |
| | Maize | 7.0 | 8.4 |
| | Pigeon pea | 6.7 | 8.3 |

3.1.4. Humid Region

In the Kacharghat village, the artificial wetland shows a very remarkable increase of 6.8 times in the CLUI value. With this intervention, the farmer is able to obtain a higher production with a 64% increase in the CPI value. At the same time, the CDI value increased by 15%, indicating diversified produce generated from the same land (Figure 17). The wetland intervention raised the value of the CFI close to 1.0 with an increase of 3.43 times compared to before the intervention. This finding indicates that the artificial wetland construction improves the fertilizer consumption in accordance with the recommended dose of fertilizer. Likewise, the increase in the CWPI value from 0.41 to 0.96 after the intervention indicates the efficient utilization of available water resources. Consequently, the gross productivity escalated by more than 200 times, and the gross return enhanced by 34% after the wetland intervention.

In the Purba Daluchhara village, the intervention of the stream-fed unlined earthen pond resulted in an increase in the overall income as well as the overall productivity of paddy and vegetables produced from the irrigated cropped area. Consequently, the values of both the CLUI and CPI increased by more than 50% after the intervention (Figure 17). The stream-fed earthen pond boosted the CDI value by 46%, with an increase of 1.2 times in the CWPI value. The CFI value shows a slight increase of 6% after the intervention, indicating that the consumption of fertilizer exceeds the recommended dose. of the SI value after the intervention is found to have an increase of 100%. It is seen that the gross productivity after the intervention is slightly reduced by 3.6%; however, the gross returns show a remarkable increase of 73%. The share of water allotment from the stream is partially used for immediate irrigation and partly harvested and stored in the earthen pond. Currently, fish contributes the most to the gross returns (40.28% of the total revenue), followed by vegetables such as spine gourd (20.32%), potato (15.69%), cauliflower (12.20%), and cabbage (8.72%).

In the Hathai Kotor village, the intervention of jalkund led to the cultivation of vegetables, arecanut, lemon, and fodder crops without any water stress even during post-monsoon season in the homestead plots located in the highlands. The intervention resulted in an increase of 1.66 times in the gross productivity and an increase of 3.8 times in the gross income. Thus, the jalkund intervention contributes to enhanced food security at the site. In addition, the values of the crop production and diversification-related indices increased after the intervention. The CLUI value indicates an increase of 22%, and of the CPI and CDI values increased by 89 and 13%, respectively, signifying a beneficial impact of

surface water harvesting on small ponds (Figure 17). Likewise, the CWPI and CFI values both record a 41 and 4% increment, respectively (Figure 17). The SI value approaches 100, which reveals the full storage of the pond filled from the additional water.

In the New Medziphema village, after 3 years of study in 2021, it is observed that the value of the CLUI increased by 15%. This finding suggests that with the intervention of jalkund, the farmers could better utilize the arable land both spatially and temporally. Similarly, the CPI value increased tremendously (52%) after the intervention. The cultivation of winter vegetables due to the availability of irrigation water plays a significant role in the 12.31% increase in the CDI value, which suggests a reduction in the risk of crop failure. Accordingly, the values of both the CWPI and CFI also show an increase of 51% and 3.12 times, respectively, compared to the values recorded before the intervention. The gross production of the crops increased by 5.46 times with a 43% increase in their gross returns (Figure 17), with an average net return of INR 3000–5000 per unit of jalkund water storage capacity.

In the Borgang village, the rearing of pigs increases monetary benefits in terms of an added animal component. It is revealed that each piglet, initially costing INR 6000, fetches about INR 15,000 after a period of six months with no involvement of any additional input costs, as the pigs are mainly fed household food wastes. The water harvested from inside the pond is used for multiple purposes, including the rearing of fish fingerlings, which further add to the gross returns. The stocking of 200 fish fingerlings weighing around 6 to 9 g results in the production of 70 kg of fish after 7 months of culture, which is sold in the market at a price of INR 300 kg$^{-1}$. After attending training at the ICAR-Research Complex for North Eastern Hill Region, Barapani, the farmer starts breeding common carp and earns an extra income of INR 40,000 annually. The various indices of farm production and diversification as a result of pond water harvesting are depicted through bar charts in Figure 17. It is seen that the value of the CLUI increased from 0.36 to 0.45, and the value of the CPI is almost double after the intervention of the water harvesting pond. Likewise, the values of the CDI and CFI show a 38 and 9% increase, respectively, after the intervention. The value of the CWPI considerably improved from 0.2 to 0.77 after the intervention. Consequently, the gross productivity records an increase of 2 times, and the gross return rose by up to 9 times. The high gross return is due to the introduction of piggeries and fisheries, which are dependent on the availability of water.

### 3.1.5. Western Ghats Mountainous Region

Before the introduction of farm ponds in Ooty, the farmers could cultivate only a single crop of potato and earn an income of INR 5.40 lakh ha$^{-1}$. After the intervention of farm ponds, the cultivation strategy was modified with the introduction of beans as the first crop under supplemental irrigation, and potato as the second irrigated crop, which increased the production level by 63%, mainly due to the added yield of beans and enhanced yield of potato. Furthermore, in high rainfall regions, the potato yield is generally reduced during monsoon season because of water-borne diseases. However, there is no scope for such diseases when potato is cultivated as the second crop. It is revealed that the introduction of farm ponds increased the values of the CLUI, CPI, and CDI by 31, 41, and 33%, respectively, while the value of the CFI decreased by 8% due to the cultivation of beans as the first crop, which enhances the nitrogen fixation (Figure 17). Overall, the intervention of farm ponds resulted in a 63% increase both in the gross productivity as well as the gross returns (Figure 17).

In Goa, about 2500 m$^3$ of water is utilized for irrigating an area of 1.21 ha for a four-month period, and as a result, the average increase of 820 and 610 kg ha$^{-1}$ in the yields of cashew and mango is obtained. In addition, 600 kg ha$^{-1}$ of cowpea yield as an intercrop is obtained. The soil of the Konkan region of Goa is laterite and highly porous in nature, which requires frequent irrigation. Hence, the increase in the yield of cashew and mango is due to supplemental irrigation given during the post-monsoon period (from November to February), which results in an increase in the flowering and seed/fruit formation. The

additional income due to cowpea production leads to a 44, 44, and 29% increase in the values of the CLUI, CPI, and CDI, respectively. The intervention of lined farm ponds results in a 29% increase in the gross productivity as well as gross returns (Figure 17).

### 3.1.6. Central or Bundelkhand Region

At the BUAT research farm, the total pumping hours during a crop year vary from 413.4 h (2021–2022) to 658.8 h (2017–2018), with the average annual pumping hours of 493.8 h during the last five years (2017–2018 to 2021–2022). The gross irrigated area varies from 29 to 42.5 ha year, with an average irrigation intensity of 232%. This indicates that a minimum of two irrigations are taken from the farm ponds. The details of the annual utilization of pond water are summarized in Table 5. Of the total storage capacity of the farm ponds, about 38.7 to 46.7% of storage water, with an average of 41.3%, could be utilized during the last five years. The average reduction in the storage capacity is found to be 2306.6 $m^3$ annually during the last five years due to undulated topography and the lack of field outlets for the safe disposal of excess runoff from agricultural lands. The storage capacity of farm ponds is reduced by up to 18.7% over a period of five years due to siltation. Therefore, the desilting of ponds is recommended at least at an interval of five years. In addition, there is a need to take appropriate soil conservation measures to restrict soil erosion from farmlands.

**Table 5.** Yearly utilization of stored rainwater for crop production.

| S. No. | Crop Year | Gross Storage (m³) | Available Storage (m³) | Net Irrigated Area (ha) | Gross Irrigated Area (ha) | Irrigation Intensity (%) | Pumping Hours | Irrigation Water Utilized (m³) | Potential Gross Irrigated Area (ha) |
|--------|-----------|--------------------|------------------------|-------------------------|---------------------------|--------------------------|---------------|--------------------------------|-------------------------------------|
| 1 | 2017–2018 | 58,468 | 40,928 | 19.0 | 42.5 | 224 | 658.8 | 19,125 | 91.0 |
| 2 | 2018–2019 | 55,776 | 39,043 | 15.3 | 35.8 | 234 | 536.7 | 16,101 | 86.8 |
| 3 | 2019–2020 | 53,468 | 37,428 | 12.0 | 29.0 | 241 | 438.0 | 14,500 | 74.9 |
| 4 | 2020–2021 | 51,545 | 36,082 | 12.5 | 29.2 | 234 | 421.9 | 14,600 | 72.2 |
| 5 | 2021–2022 | 50,007 | 35,005 | 13.0 | 29.5 | 227 | 413.4 | 13,880 | 74.5 |

At present, an area of about 20–40 ha of the BUAT research farm under kharif or rainy season crops (pigeon pea and sesame) and rabi or winter season crops (chickpea, mustard, green gram, and wheat) has been irrigated at least twice through the harvested water in farm ponds since 2016–2017. In general, the ponds are filled to the maximum storage capacity on the occurrence of 600 to 1000 mm of annual rainfall. However, only 70% of the pond water storage capacity was filled in 2017. The unit cost of harvested rainwater for the 30-year average pond life comes to be INR 1.48 $m^3$ for the full storage capacity, which increases to INR 2.11 $m^3$ for 70% of the total capacity in below average rainfall years. Hence, the construction of farm ponds is economically viable with a satisfactory unit cost of the stored rainwater in the Bundelkhand region of the country. Moreover, water harvesting through farm ponds has a benefit–cost ratio of 2.23, a net present value of INR 9,745,402, and an internal rate of return of 18.1% for the productive life period of 30 years of the water harvesting ponds, which are the only source of irrigation in semi-arid regions of the country (Table 6).

### 3.2. Statistical Significance of Water Harvesting Intervention

In this study, the change in the specific values of the individual crop production and diversification indices, i.e., CLUI, CPI, CDI, and CWPI, along with the GP and GR before and after the implementation of the water harvesting structures, are statistically evaluated using the paired-sample Wilcoxon signed-rank test. The significance of all the indices is assessed at $p < 0.05$. The results of the test in terms of the $p$-value are shown on bar charts of the crop production and diversification indices (Figure 17). It is seen that the $p$-value of all

the indices, except for the CFI, are less than 0.05, which indicates that there is a statistically significant difference in the values of the CLUI, CPI, CDI, CWPI, GP, and GR before and after the intervention of water harvesting. This finding clearly reveals the significant impact of different water harvesting interventions on crop production and diversification indices throughout the country. Therefore, it is recommended to adopt water harvesting through location-specific means in different climate settings of the country to enhance agricultural production and improve food security of the nation.

**Table 6.** Economic evaluation of farm ponds for the duration of 30 years.

| Item | Irrigation through Farm Ponds Only |
| --- | --- |
| Present worth of incremental return (INR) | 17,680,012 |
| Present worth of total cost @ 10% (INR) | 7,934,610 |
| Benefit–cost ratio | 2.23 |
| Net present value (INR) | 9,745,402 |
| Internal rate of return (%) | 18.1 |

*3.3. Up-Scaling Potential*

The total annual precipitation of India is about 4000 billion cubic meters (bcm), and out of that, around 1150 bcm of rainwater escapes from the land and goes to the seas without utilization in agriculture due to high-intensity storms [41]. It is well understood from the findings of the 12 case studies presented in this article that there is a vast scope to capture the surplus/excess runoff through cost-effective water harvesting ponds, including jalkund and a dug well attached with a recharge filter, at feasible locations and in adequate amounts in the upstream landscape. However, the impact of harvesting the entire surplus/excess water in the upstream area may have concerns with the water availability in the downstream area, and hence, there is a need to balance enhancing the potential of water harvesting toward upstream landscapes without impacting the environment or landscape hydrology toward the downstream landscapes. This study explores only the scope of the up-scaling potential of the water harvesting interventions, and the impact may be evaluated in a future study. By adopting this technology, the cropping intensity, crop productivity, and water use efficiency could be enhanced. The up-scaling potential of the rainwater harvesting technique in six regions of the country is summarized in Table 7, and the same topic is explained and succinctly discussed ahead.

3.3.1. Northwestern Himalayan Region

According to the estimates of the Statistical Abstract [42], an area of about 135,889 ha in Himachal Pradesh is under single cropping due to the limited availability of irrigation water. Thus, water flowing through the springs in the hilly areas of the Himalayan region can be collected and used for providing supplemental irrigation, which can bring the land under single cropping area to double cropping. It is further estimated that this conversion of the land from single cropping to double cropping may result in a significant increase in the gross production of the entire Himachal Pradesh by up to $129 \times 10^4$ tonnes, transforming the gross returns up to INR 1888.85 crores through supplemental irrigation.

In Uttarakhand, only 14% of the total reported area (53,483 km$^2$) is under cultivation, and more than 55% of the cultivated land is rainfed. The cropping intensity in the state is 160.6%, with the average land holding 0.68 ha (that, too, is divided into many patches) in the hills and 1.77 ha in the plains [43]. In the state, 412,407 ha of the cropped area is rainfed, of which about 10% can be brought under two crops by providing supplemental irrigation through rainwater stored in ponds/tanks, which could transform the rainfed cropped land to increase food production and provide livelihood security.

**Table 7.** Up-scaling potential of water harvesting intervention in six regions of India.

| Region | Water Harvesting Intervention | Current Status | Up-Scaling Potential | Advantages |
|---|---|---|---|---|
| Northwestern Himalayan Region | Spring water storage in tanks and subsequent utilization for supplemental irrigation. | An area of 135,889 ha in Himachal Pradesh is under single cropping. | • In Himachal Pradesh, gross crop production may reach up to $129 \times 10^4$ tons with gross returns of up to INR 1888.85 crores.<br>• In Uttarakhand, 10% of rainfed cropped area (412,407 ha) can be brought under double cropping. | • Provision of supplemental irrigation<br>• Single cropping to double cropping<br>• Increase in crop production |
| Western Hot Arid Region | Rainwater harvesting pond for supplemental irrigation to rabi season crops. | Low agricultural productivity in 32 million ha hot arid lands; major part in Rajasthan and Gujarat. | • It is expected that 10% area (3.2 million ha) of total hot arid lands in the country may be brought under farm pond irrigation. | • Additional cropland under irrigation<br>• Enhanced income from agriculture<br>• Efficient rainwater management<br>• Improved agricultural productivity |
| Western Semi-Arid Region | Jalkund (small plastic-lined water harvesting structure) and recharge filter attached with dug well. | In Gujarat, total cropped area in 12.8 million ha and net sown area in 10.1 million ha. However, only 34.4% of area is under irrigation. | • About 15% of net sown area can be brought under irrigation through water harvesting interventions, and thus, the irrigated area would be 50% instead of 34.4%. | • Double cropping intensity and gross revenue<br>• Reclamation of alkaline and saline soils<br>• Bringing more areas under irrigation |
| Humid Region | Artificial wetland for aquatic plants; stream-fed multipurpose pond; plastic-lined rainfed pond. | In Nagaland, net cropped area = $385 \times 10^3$ ha; 30% is irrigated. Total area in northeast India = 262,179 km$^2$; forest land = 169,521 km$^2$. | • In Nagaland, it is expected that irrigated area may be increased by up to 50% of the net cropped area by covering an additional $77 \times 10^3$ ha.<br>• In northeast India, it is expected that 10% more area will be under supplemental irrigation, with 9265 km$^2$ area with assured irrigation throughout the year. | • More areas under irrigation<br>• Assured irrigation throughout the year |

**Table 7.** *Cont.*

| Region | Water Harvesting Intervention | Current Status | Up-Scaling Potential | Advantages |
|---|---|---|---|---|
| Western Ghats Mountainous Region | Earthen and plastic-lined rainwater harvesting pond. | Total area is 160,000 km$^2$ in a stretch of 1600 km parallel to Indian west coast. | • About 10% of the area in Western Ghats region (i.e., 16,000 km$^2$) can be brought under assured irrigation through farm pond intervention. | • More cropland areas under assured irrigation<br>• Better livelihood for the farmers of the region |
| Bundelkhand or Central Region | Dugout-cum-embankment type of farm pond. | Total irrigated area is 13.7 × 10$^6$ ha, which is 66% of the total sown area. | • Dugout-cum-embankment farm pond may be up-scaled in additional 10% of the total sown area through flagship scheme of the Indian government being implemented in the region. | • Assured irrigation to rabi season crops in the water scarce region<br>• Eradicating problem of water crisis |

### 3.3.2. Western Hot Arid Region

In the research farm of CAZRI in the Kachchh district, the efforts of rainwater harvesting result in additional land to be brought under irrigated croplands, which helps farmers obtain additional income from agriculture. Thus, it is obvious that the efficient management of harvested water has the potential to improve agricultural productivity in the hot arid Indian lands, extending over an area of 32 million ha, with a major portion in Rajasthan and Gujarat [44]. With the expectation that 10% of the area will be covered under farm pond irrigation, it is expected that an area of about 3.2 million ha of the hot arid lands of the country may be brought under irrigation from suitable interventions of rainwater harvesting to provide supplemental irrigation to arid crops.

### 3.3.3. Western Semi-Arid Region

Gujarat has 10.1 million ha of net sown area and 12.8 million ha of total cropped area, of which only 34.4% of the area is irrigated. Almost 65–70% of the total area is rainfed and drought prone. Thus, about 15% of the net sown area could be brought under irrigation through the interventions of jalkund and a recharge filter attached with a dug well. It could be expected that after adopting this technology, the irrigated area of Gujarat would be 50% instead of 34.4%. Similarly, the cropping intensity and gross revenue of the farmers would double. The state has a huge area under alkaline and saline soils, which would be effectively reclaimed by bringing more area under irrigation through the construction of the water harvesting structures.

### 3.3.4. Humid Region

Nagaland has a total geographical area of 16,579 km$^2$. The state has a total surface water potential of 19,483 million m$^3$ [45]. The net cropped area of the state is estimated to be $385 \times 10^3$ ha, while the irrigated area is only 30% of the net cropped area. After the adoption of the rainwater harvesting intervention, it is expected that the irrigated area may be increased by up to 50% of the net cropped area by covering an additional $77 \times 10^3$ ha of the net cropped area under irrigation through the efficient utilization of the harvested rainwater. To achieve this target, about 18.20 lakh numbers of small-sized (5 m $\times$ 4 m $\times$ 1.5 m) water harvesting structures, i.e., jalkund, will need to be developed. The small-sized jalkund is found to be suitable for the hilly areas; however, the medium- and large-sized ponds may also be used particularly to irrigate crops in the foothill areas.

The northeast region of the country covers 262,179 km$^2$ area, of which about 169,521 km$^2$ of the area is under forestland [46], where no agriculture activity may take place following the legislations implemented to protect the forests in the country. After excluding the forest lands, the remaining area of 92,658 km$^2$ has the scope for implementing adequate rainwater harvesting interventions. In the region, the average annual rainfall is 2000 mm, with variations ranging from 1500 to 12,000 mm [47]. It is expected that 10% of the area where supplemental irrigations may be provided through the suitably constructed farm ponds, and a total area of 9265 km$^2$ can be covered with assured irrigation throughout the year.

### 3.3.5. Western Ghats Mountainous Region

The Western Ghats mountain ranges of the country cover an area of 160,000 km$^2$ in a stretch of 1600 km parallel to the west coast of the Indian peninsula. The Western Ghats region receives the rainfall ranging between 1200 and 3800 mm, with the maximum rainfall received during the monsoon season (June through September). About 10% of the area in the Western Ghats region (i.e., 16,000 km$^2$) can be brought under assured irrigation through farm pond intervention, which would translate into better livelihood for the farmers of the region.

### 3.3.6. Bundelkhand or Central Region

The concept of water harvesting, developed and employed at the research farm of BUAT at Banda, can be replicated in other parts of the area through the development of a community-based or individual farm-scale rainwater harvesting system. Furthermore, harvested water may be utilized for providing life-saving irrigation during the kharif season and the assured irrigation to the rabi season crops in the water scarce region of Bundelkhand. The groundwater is geo-morphologically controlled in the region, and hence, a series of farm ponds may have a significant impact to eradicate the problem of water crisis in the area. The concept of dugout-cum-embankment type farm ponds may further be considered for up-scaling through a special package or flagship scheme of the Indian government, which is being implemented in the region.

## 4. Discussion

Water is a vital component of the hydrologic cycle which assures food security and determines the full potential of the agriculture sector in a country. Indian agriculture is largely rainfed, and hence, water harvesting is essentially required to explore the fullest potential of the agricultural lands. Large water harvesting structures such as dams are not able to fulfill the requirement of far flung and hilly areas, and in such situations, small ponds have better potential both in terms of cost effectiveness and flexibility in choosing the location and dimension. Singh et al. [48] concluded that subsistence agriculture in the hilly region could be successfully transformed into a profit-earning enterprise by tapping and utilizing water resources. Small-sized ponds are suitable for both high rainfall areas as well as low rainfall areas. In different locations of the country, it was found that the irrigation through farm ponds increased the yield and also resulted in increased crop diversification. Such yield increment patterns due to the intervention of farm ponds are reported in many studies, e.g., Omar Munyaneza et al. [49] and Reddy et al. [9], among others. The farm pond irrigation further enhanced agricultural productivity in Indian scenarios as reported in the literature [50,51]. The introduction of high-yielding variety, which is generally more water demanding, is possible only when adequate water quantities are available. Rainwater harvesting through farm ponds offers a valuable tool to store a sufficient amount of rainwater for its subsequent utilization in raising crops, which otherwise would not be available for agricultural production after cessation of the rainfall. Kumawat et al. [11] observed that the high-yielding varieties of the food and fruit crops achieve their maximum yield potential when irrigated in a timely manner by utilizing water harvested from farm ponds and following an adequate irrigation schedule. Similarly, Sen et al. [52] reports an increase in the number of crops grown and their duration in the field due to the increased availability of irrigation water through water harvesting structures, which results in better values of economic parameters. Farm ponds in the northeastern region of the country are considered as the most remunerative option in the agriculture sector as more than 90% of the population in the area are fish eaters, and hence, a significant increase in the gross revenue returns from agricultural products is recorded.

Naik et al. [53] implements micro-sized water harvesting systems, i.e., jalkunds of 3 m× 1.5 m × 1 m in size, in Central India. It is reported that the water harvesting technology is capable of sustaining 10 plants of mango fruit for 6 months. Thus, it can be seen that such a small water harvesting structure can be dug in agricultural fields at small intervals to establish fruit orchards. The benefit–cost ratio of 2.3:1 is achieved in the sixth year of implementation of the jalkunds technology due to the 100% survival of fruit plants. Previously, in conventional irrigation systems, fruit plants are damaged, with a 50% mortality of the orchard plants due to the absence of assured water supplies, and the agriculture enterprise is not profitable, as evidenced from the benefit–cost ratio of 0.87:1.

While reviving kitchen gardening and other agricultural activities in the backyard, a low-cost technology for water harvesting structure, i.e., jalkund, is constructed in different villages of the Wokha district, Nagaland. The crops grown and livestock raised in different villages include vegetables such as cabbage, tomato, coriander, fish stockings, nursery

raising, and pig rearing. It is reported that the cropped area under irrigation from jalkund varies from 240.5 m$^2$ to 700 m$^2$, and the irrigation water supplies vary from 27,450 to 111,229 L. The gross returns from a unit area with the intervention of jalkund range from INR 17,550 to 68,900 ha, with the mean value of the benefit–cost ratio ranging from 1.73 to 2.53 [54].

Ray et al. [55] report that the provision of irrigation water due to the construction of water harvesting structures in the integrated farming system provides an opportunity for the double cropping in the northeastern hill region of the country. The introduction of winter crops in the integrated farming system increases the cropping intensity by 162.6–178.9% at the intervention sites over the traditional method of shifting cultivation (100%).

## 5. Conclusions

In India, climatic conditions vary widely from arid to humid tropics, with the annual rainfall ranging from 400 to 11,000 mm. About 70% of the population is engaged in agriculture that is predominantly rainfed. Even in high rainfall regions, water scarcity is experienced both before and after the monsoon, as about 70 to 80% of the rainfall is received only during the monsoon season (July–September). Groundwater extraction for irrigated agriculture is expensive as well as unsustainable because it exceeds the amount of natural recharge. Thus, rainwater harvesting-cum-supplemental irrigation is a viable option to sustain agriculture even under the impact of climate change, which causes irregular onsets/withdrawals, long dry spells, high-intensity storms, and unseasonal rains. A total of 12 case studies from different climatic conditions of the country are presented in this paper, which reveal that water harvesting through farm ponds is fruitful in enhancing crop diversification index, fertilization index, land utilization index, and income enhancement. The average increase in the value of the cultivated land utilization index signifies that the cultivated fallow lands are better utilized either by bringing them under cultivation or by maintaining crop cover for a longer duration. Additionally, areas with single crop start producing a higher number of crops in a year after the intervention of rainwater harvesting. All sorts of water harvesting lead to assured water availability for irrigation purposes, and as a result, farmers tend to cultivate diverse crops that ensure certain income to farmers by reducing the risk of complete crop failure mostly prevailing under the mono-cropping system. Furthermore, the farm pond intervention provides an assured water supply, and thus, increases the value of the crop diversification index and crop productivity index. In India, only 39.55% of the total available agricultural land is irrigated. Targeting only 5% of the rainfed agricultural land of India with irrigation from farm ponds will bring an additional 9044 × 10$^3$ ha area under irrigation. Moreover, it is evident that the water harvesting intervention in all the case studies causes statistically significant ($p$-value < 0.05) improvements in the value of crop production and diversification indices. Based on the average crop productivity index value of the case studies, it is evidenced that with farm pond intervention alone, the productivity of the rainfed areas will significantly increase and consequently, about 4.40 million Mg of additional food grain production can be realized. Moreover, a remarkable step in this direction has already been taken by the government of Haryana, which is one of the agricultural states of the country, by focusing on the construction of farm ponds at a large scale by creating the "Farm Pond Authority" in the state. It is further emphasized that the construction of water harvesting ponds should be accomplished at only feasible locations and appropriate proportions so as not to impact the environmental flow.

**Author Contributions:** Validation, V.K. (Vandita Kumari); investigation, P.P., D.M., S.M., P.R.B., J.M.S.T., R.K., D.J., P.K.S., L.K.B., N.P.D., S.K. (Sanjay Kumar), P.D., V.K. (Vijaysinha Kakade), G.S., N.R.S., S.G.S., A.P., P.S.R., S.P., V.K.B., N.K.S., O.P.S.K., S.K.R., V.K.T., P.L.B., K.N., R.D., D.D., A.K., G.S.P., S.V.D., S.K. (Sanjeev Kumar), and B.K.S.; writing—original draft, P.P.; writing—review and editing, D.M. All authors have read and agreed to the published version of the manuscript.

**Funding:** This research was funded by the Indian Council of Agricultural Research, New Delhi in some of the case studies.

**Institutional Review Board Statement:** Not applicable.

**Informed Consent Statement:** Not applicable.

**Data Availability Statement:** Not applicable.

**Acknowledgments:** The authors duly acknowledge the resources and necessary facilities provided by the directors of the research institutes and vice-chancellors of the universities involved in the study. Further, the authors are grateful to the three anonymous referees for their meticulous comments and useful suggestions that helped improve the earlier version of the manuscript.

**Conflicts of Interest:** The authors declare no conflict of interest.

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
