# Peer review of "Sustainable Water Harvesting for Improving Food Security and Livelihoods of Smallholders under Different Climatic Conditions of India"

_sustainability, doi:10.3390/su15129230_

Round 1

Reviewer 1 Report

This paper provides an interesting overview of the challenges presented to Indian agriculture by a drying climate. Further, it highlights the opportunities that arise through the implementation of 'low tech' mitigation responses that enhance insitu water management and enterprise diversification.  That being said I have some reservations about the paper in its current form. 

Firstly, the paper would benefit from professional editing to ensure that the content was clear and concise.

The paper would benefit from additional editing to consolidate/remove extraneous materials that detract from the thrust of the narrative, which is the benefit accruing as a result of the harnessing of scarce resources and the subsequent improvement in land usage and business diversification. Specifically I refer to section 2,2 (Lines 286-506), which provides excessive detail related to the rationale and construction of storage.  While this material is of interest it is not germane to the focus of the narrative and could be adequately accommodated in much summarised form in Section 2.1.  

The discussion relating to the indicators relied apon for water harvesting performance are referenced to documents that are not widely available *eg Shardi et al 2012).  While the indicators are intuitively applicable, the paper would benefit from referencing these performance indicators to internationally recognised sources to enable a critical assessment by the reader. Note that Brooks et al 1998 and Yuan et al 2003 are not recognised in the reference list.

A key result for this paper is that the availability of water enabled the adoption of new farming systems, which in turn generated more revenue and on analysis higher net returns. It would be useful for the reader to have the changes in production activities summarised in a table rather than embedded in the text as it currently is. On a similar note, more attention is required with respect to the differences in costs between farming systems before and after access to more secure water resources.

Economic analyses use the term - Present worth of incremental return - that is not anticipated in the earlier performance indicators and should be clarified.

Section 3.3 considers the upscaling of this approach. There is implied in this that water that is not harvested is wasted.  This is a narrow construction of the value of water in the broader environment in which agriculture is situated. Your discussion would benefit from consideration of what large scale uptake of the approach might mean for human ad environmental outcomes.  There is ample evidence from other parts of the world (eg Australia) that an upswing in teh prevalence of small farm dams has a significant impact on broader catchment outcomes.

Reviewer 2 Report

Figure 1. It is not map, because the absence of information: 1) coordinate system, 2) legend, 3) wind direction, 4) scale. Author may write the climate zonation in India. Marker of study area must be change to other color such red. 

Table 2 must be shown by graph. It is more informative 

Sub heading 3.3. Please present in graphic/table

Sub heading 3.1. Crop production is not solely influenced by water or climate. The management such as tillage, fertilizer, pest control are vital. How do authors address this issue in the study? Because we could not find the information of management of agriculture at the 6 pilot project. 

Conclusion. It should be concise and answer the purpose of study and no citation here (line 959). Please not discussion here!

Reviewer 3 Report

Dear Authors
the article represents the report of very interesting agronomical project -
though its lacks of an orthodox scientific approach it is
an attractive and instructive reading - minor revision are needed, while every authors
are asked to reread the article.

Title - though most of readers should know about the agrcultutre in India, could benefit evidence that the paper focus smallholders and peasant agriculture

Unit of measure: define Rs=Rupias (just the first time) and avoid other symbols (Table 4)
- check for commas everywhere (L47, L140, L611, Table 4)
- se always mm (L127)

Some paragraphs should be revised: L69-70, L77-78, L95-96, L128-130, L165-168

Check for typos, e.g.
L140: ghat -> Ghat
L143: month -> months
L193: 'in 2°C" -> "is 2°C"
L309 "number of" remove
L515 "water storage" -> "dedicated" (avoid repetitions)
L663 "increased" -> "higher" (avoid repetitions)
L674 there is a space before the end of sentence

Method:
Please check climate in Tripura sites: why in a "Humid Region" you have in 381mm (L233) and 200mm (L243) only ?
Table 1: can't find the definition of CI index in the text
Add in parenthesis BCR, NPV and IRR after their listing (L545)
use T = Minimum(item1, item2)    (574)

L580-584 - report this in method (above) or conclusions

Table 2 - put in L590
Table 2 - consider to transpose to make it more readable

Round 2

Reviewer 1 Report

This paper provides an interesting overview of the varied challenges faced by rainfed agriculture in various geographic environments in India.  it provides an interesting review of 'low tech' interventions to enhance production through the storage and addition of water at key stages in the production cycle.  I acknowledge the work that has been done in addressing earlier concerns. Some key points remaining are:

Referencing:

Further editing is required to ensure that all assertions are supported by evidence. It is not uncommon for blocks of text to be unsupported.  See for example Introduction P1 L1-L11 or P2 L7-L15

Value-judgements with respect to water:

The focus of this paper is on harnessing water for irrigation, however, that does not mean that water in the landscape that flows past as runoff is 'wasted'.  More attention needs to be paid to the limitation of judgmental language such as this such that the paper recognizes the value of water in the landscape, one element of which is for agricultural production. See for example Introduction P1L15, ie "The rainfall is received during the monsoon season, and a significant portion of it is lost as run off."

Readability:

The paper would benefit from additional editing to address flow of ideas. Specifically, this refers to grammatical editing.

Structure and Content:

Notwithstanding the response to earlier reviewer feedback, Sections 2.1 and 2.2 would be better consolidated to reduce the volume of text and provide for a more concise understanding of the case studies.

Exploration of impacts of upscaling:

The paper considers the potential for upscaling the 'low-tech' interventions. However, the paper does not consider the impacts of this intervention in water flows on the broader environment. It would be beneficial to understand how much water would be captured as a consequence of the upscaling, expressed as a proportion of total water in the landscape. What impact will broadscale uptake of the intervention have on availability for other water users? This is the crux of questions relating to the sharing of a scarce resource such as water and is 'live' in irrigation communities around the world impacted by climate change.  While a deep investigation of this issue may be considered beyond the scope of this particular work, it is important that the magnitude of the proposed intervention at scale is understood.
